



# Experimental investigation of Mini Gurney Flaps in combination with vortex generators for improved wind turbine blade performance

Jörg Alber[1], Marinos Manolesos[2], Guido Weinzierl-Dlugosch[3], Johannes Fischer[3], Alexander Schönmeier[1], Christian Navid Nayeri[1], Christian Oliver Paschereit[1], Joachim Twele[4], Jens Fortmann[4], Pier Francesco Melani[5], Alessandro Bianchini[5].

[1] Technische Universität Berlin, Hermann-Föttinger Institut, Müller-Breslau-Str. 8, 10623 Berlin, Germany
[2] College of Engineering, Swansea University, Bay Campus, Fabian Way, Swansea, SA1 8EN, United Kingdom
[3] SMART BLADE GmbH ®, Waldemarstr. 39, 10999 Berlin, Germany
[4] Hochschule für Technik und Wirtschaft Berlin, Wilhelminenhofstraße 75A, 12459 Berlin, Germany
[5] Università degli Studi di Firenze, Department of Industrial Engineering (DIEF), Via di Santa Marta 3, 50139 Firenze, Italy

*Correspondence to*: Jörg Alber (joerg.alber@posteo.de)

## Abstract

This wind tunnel study investigates the aerodynamic effects of Mini Gurney flaps (MGFs) and their combination with vortex generators (VGs) on the performance of airfoils and wind turbine rotor blades. VGs are installed on the suction side aiming at stall delay and increased maximum lift. MGFs are thin angle profiles that are attached at the trailing edge in order to increase lift at pre-stall operation. The implementation of both these passive flow control devices is accompanied by a certain drag penalty. The wind tunnel tests are conducted at the Hermann- Föttinger Institut of the Technische Universität Berlin. Lift is determined with a force balance and drag with a wake rake for static angles of attack from -5° to 17° at a constant Reynolds number of 1.5 million. The impact of different MGF heights including 0.25%, 0.5% and 1.0 % and an uniform VG height of 1.1 % of the chord length are tested on three airfoils that are characteristic for different sections of large rotor blades. Furthermore, the clean and the tripped baseline cases are considered. In the latter, leading edge transition is forced by means of Zig Zag (ZZ) turbulator tape. The preferred configurations are the smallest MGF on the NACA63(3)618 and the AH93W174 (mid to tip blade region) and the medium sized MGF combined with VGs on the DU97W300 (root to mid region). Next, the experimental lift and drag polar data is imported into the software QBlade in order to design a generic rotor blade. The blade performance is simulated with and without the add-ons based on two case studies. In the first case, the retrofit application on an existing blade mitigates the adverse effects of the ZZ tape. Stall is delayed and the aerodynamic efficiency is partly recovered leading to an improvement of the power curve. In the second case, the new design application allows for the design of a more slender blade while maintaining the power output. Moreover, the alternative blade appears to be more resistant against forced leading edge transition.





## 1. Introduction

### 1.1 General outline

The study is divided into the following sections.

1. **Introduction.** The concepts, mechanisms and applications of Gurney flaps (GFs), ZZ tape and VGs are introduced.
The literature review is focused on very small GF heights, so-called MGFs, and the combination of GFs with VGs.

2. **Airfoil simulations.** The simulation software XFOIL (Drela, 1989) is used to determine the appropriate dimensions of each PFC device in relation to the local boundary layer thickness of the following airfoils: the NACA63(3)618, the AH93W174 and the DU97W300.

3. **Experimental set-up**. The wind tunnel test section, the measurement methods and the data reduction process are
specified including the force balance for the lift, and the wake rake for the drag measurements at a constant Reynolds number of $Re = 1.5 \cdot 10^6$.

4. **Experimental results.** The baseline lift and drag polars, $c_l(\alpha)$ and $c_d(\alpha)$, of each airfoil are validated against literature data. Different combinations of MGFs and VGs are assessed according to characteristic parameters, i.e. the lift performance, the stall behavior and the aerodynamic efficiency.

5. **Rotor blade performance.** The experimental data of the preferred configurations is imported into the software QBlade (Marten, 2020) to create a generic rotor blade. The blade performance is simulated based on two case studies, the retrofit application on an existing, and the new design application on an alternative rotor blade.

### 1.2 Gurney flaps

This aerodynamic device is named after the US racecar driver Dan Gurney. In the early 1970s, he applied it to the rear spoilers
achieving significant improvements in the downforce and thus the traction of his Formula One vehicles, see Liebeck (1978). Passive GFs are categorized as static miniflaps or miniature trailing edge devices (MiniTEDs), as described by González-Salcedo et al. (2020). Hence, they are different to the concept of flexible trailing edge (TE) flaps that are integrated into the very TE section, see Barlas and van Kuik (2010). The first reference to miniflaps dates back to the early 20[th] century and was probably developed by Gruschwitz and Schrenk (1933). Zaparka (1935) registered the first patent on active miniflaps for use
on airplane wings. Various patents of passive miniflaps followed, particularly in aviation. Boyd (1984) and later Brink (2002) claimed the rights on different versions of wedge-shaped TE flaps. Henne and Gregg (1989) patented the shape of a diverging trailing edge (DTE) of a transonic airfoil generating similar aerodynamic effects than the GF. Bechert et al. (2001) registered a patent on so-called three dimensional (3D) GFs, i.e. profiles with slits, serrations, holes, as well as tiny vortex generators attached to the miniflap itself in order to stabilize the otherwise unsteady wake field. Wang et al. (2008) published a
comprehensive review of GFs for use on rotor blades of helicopters and wind turbines. In contrast to the large amount of patents and publications, there are only few examples of standardized or commercialized GF applications on rotor blades of horizontal axis wind turbines (HAWTs). For instance, Vestas (2019) offAers GFs in combination with VGs as aerodynamic





upgrades of operating wind turbines predicting the average energy production (AEP) to increase by 1.7%. Another example is the blade design of the 10 MW reference wind turbine of the Danish Technical University (DTU) with a total rotor radius of

$R = 89.2$ m. The inner blade part alongside the local rotor radius of $5\%R < r < 40\%R$ was equipped with wedge-shaped GFs including heights of 3.5%, 2.5% and 1.3%, respectively, in relation to the local chord length. Bak et al. (2013) claim significant aerodynamic performance improvements, especially on relatively thick airfoils with a maximum thickness of $h_{th,max} \geq 30\%c$.

(a)                                                                        (b)

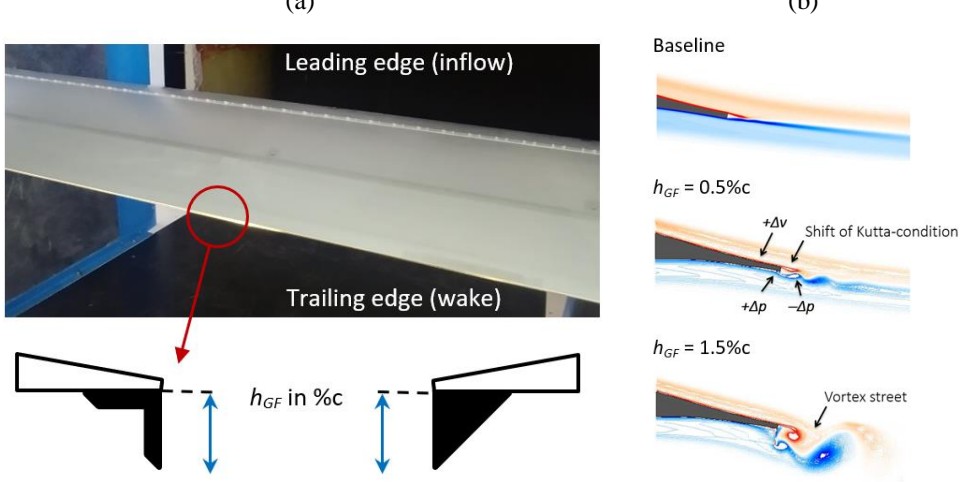

Figure 1. (a) NACA63(3)618 during wind tunnel tests. Vortex generator array and Gurney flap. Definition of Gurney flap height of rectangular and triangular profiles in side view. (b) CFD simulations of the HQ17. Wake structures at $\alpha = 0.0°$ and $Re = 1\cdot10^6$ for different
Gurney flap heights, reproduced and modified from Schatz et al. (2004a).

Figure 1a displays typical GFs, i.e. the rectangular, or L-shaped, and the triangular, or wedge-shaped, profiles. They are attached at the TE of wings and rotor blades, normal to the pressure side. In both cases, the effective GF height, $h_{GF}$, is expressed in percentage of the chord length, $\%c$, without taking the original TE thickness, $h_{TE}$, into account. For identical $h_{GF}$, the aerodynamic effect of both GF profiles is considered to be very similar, as discussed in Sect 4.1.2. Figure 1b illustrates the
principal changes of the flow field for two different GF heights, as first reported by Liebeck (1978). Adjacent to the TE modification, a highly efficient vortex system is formed consisting of one vortex upstream and two counter rotating vortices immediately downstream. Bechert et al. (2000) and Schatz et al. (2004b) showed by means of experimental and numerical investigations that the wake flow structures are quasi two dimensional (2D) at pre stall operation. The recirculation region changes the Kutta condition, so that the rear stagnation point is shifted downstream and deflected downwards, see Jeffrey et
al. (2000) and Cole et al. (2013). The modifications of the flow field lead to the following set of simultaneous effects:

- Lift performance: The suction peak is higher and coupled with a positive pressure built-up right in front of the GF, increasing the pressure difference between suction side (SuS) and pressure side (PS). As a results, the effective camber is enhanced, thus shifting the lift curve upwards so that the same $c_l(\alpha)$ is already reached at a lower angle of attack (AoA), $\alpha$. Furthermore, the adverse pressure gradient on the SuS becomes milder generating higher maximum lift
85        coefficients, $c_{l,max}$, as stall is initiated.



- Drag behavior: The recirculation or low pressure region in the immediate wake causes an increased momentum loss and thus higher drag coefficients, $c_d(\alpha)$. In addition, the intensity of the unsteadiness is stronger, especially if vortex shedding is initiated in the form of an absolute instability, as illustrated in Figure 1b for $h_{GF} = 1.5\%c$.

Overall, the impact of GFs is quantifiable as an increase in both lift, $\Delta c_l$, and drag, $\Delta c_d$. Bechert et al. (2000) and Schatz et al. (2004b) showed that $\Delta c_d(\alpha)$ is directly related to the lift fluctuations, i.e. the intensity and the frequency of the wake unsteadiness. As a consequence, the drag penalty proved to be less severe for small GF heights, comparing $h_{GF} = 0.5\%c$ to 1.5%c in Figure 1b. Furthermore, in case of $h_{GF} = 0.5\%c$, the unsteadiness vanished as the AoA is increased from $\alpha = 0.0°$ to 4.0°, as illustrated comparing Figure 1b to Figure 5c. In summary, the reviewed results indicate the possibility to improve, or
at least to maintain, the lift to drag (L/D) ratio by installing rather small GFs heights in the order of $h_{GF} \leq 0.5\%c$ at $Re \approx 1 \cdot 10^6$.

### 1.3 Zig Zag tape

ZZ turbulator tape is implemented to initiate the boundary layer (BL) transition at a fixed chord position, see Figure 3a. Its height, $h_{ZZ}$, should be slightly smaller than the local laminar BL thickness in order to trigger transition while avoiding a disproportionate drag increase or even turbulent separation. Next to trip wire or carborundum paper, ZZ tape facilitates the
comparability between different measurement methods. Moreover, it is applied to evaluate the sensitivity of airfoils to adverse leading edge roughness (LER) effects, as discussed by van Rooij and Timmer (2003), Timmer and Schaffarczyk, (2004) and in greater detail by Wilcox et al. (2017). Another example is Oerlemans et al. (2009), who implemented ZZ tape on the rotor blades of research wind turbines. In fact, LER due to erosion and the accumulation of sediments are major challenges for rotor blade manufacturers and wind turbine operators, see Figure 2b. According to Maniaci (2020), LER mainly affects the mid to
tip region where the rotor blade is exposed to high relative velocities. Depending on the severity, or the degree of roughness, the AEP decrease of multi MW HAWTs is between of 2 % and 5 %.

### 1.4 Vortex generators

As opposed to GFs, VGs have been commercialized by various wind energy companies for almost two decades. The suboptimal or declining AEP over the years is often the reason for blade manufacturers and wind park operators to investigate
the possible causes, such as early separation or LER, and to invest in feasible solutions, such as PFC devices. VGs are implemented in order to prevent, or at least to alleviate, the flow separation in the root to mid region of rotor blades. More recent studies have investigated the opportunities of relatively small VG heights in the mid to tip blade region, see Bak et al. (2018). According to Lin (2002), they are referred to as sub- BL or low profile VG configurations, see Sect. 2.5. Typically, VGs are commercialized as retrofit solutions, i.e. add-ons that are installed onto the surface of already running rotor blades, as
depicted in Figure 2a. In this way, SMART BLADE (2021) predict an AEP growth of approximately 2%. A more detailed review on VGs for use on rotor blades is provided by Bak et al. (2016) and González-Salcedo et al. (2020).



(a)                                                                  (b)

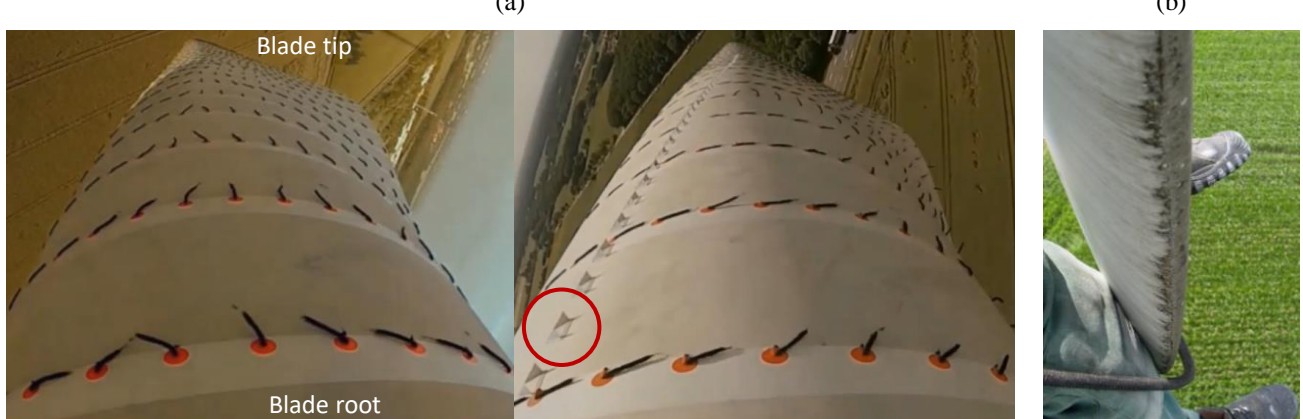

Figure 2. Utility scale wind turbines. (a) Simultaneous flow tuft measurements on different blades of the same rotor, baseline versus VG configuration, with permission from SMART BLADE GmbH. (b) Leading edge erosion at the blade tip, with permission from Seilpartner GmbH.

The purpose of VGs is to delay the stagnation of the BL flow and thus separation. Installing VGs, Figure 2a illustrates that the flow tufts are attached to the blade surface until closer to the TE, as compared to the stalling baseline blade. The thin vane tips shed a pair of vortices in order to transport momentum from the free flow into the viscous BL close to the surface. The vortex system spreads out towards the TE, where it is released into the immediate wake. More detailed research on the underlying mechanism of VGs is provided by Manolesos and Voutsinas (2015). Overall, the VG effect is quantifiable as a substantial

increase in both maximum lift, $c_{l,max}$, and the AoA where stall is initiated, $\alpha(c_{l,max}) = \alpha_{cl,max}$. However, drag is increased significantly at low and moderate AoA. The impact on the aerodynamic efficiency, $L/D$ $(\alpha)$, depends on the design parameters illustrated in Figure 3.

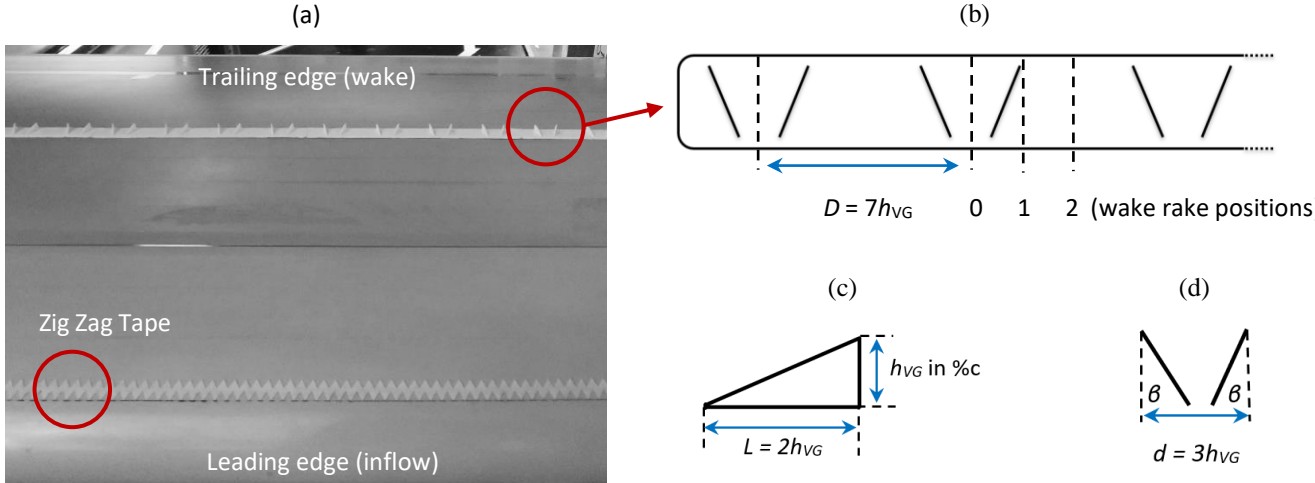

Figure 3. NACA63(3)618 during wind tunnel tests. (a) Top view on suction side with Zig Zag tape and VG array. (b) VG panel including spacing between VG centers and measurement positions of the wake rake, with permission from SMART BLADE GmbH. (c) Side view of 130  single vane. (d) Top view of pair of vanes.



Figure 3a and b depict an array of VG panels as it is installed on the airfoil SuS throughout the wind tunnel tests, see Sect. 3 and 4. Following Timmer and van Rooij (2003) and Baldacchino et al. (2018), the design parameters generate a counterrotating, common downflow VG system. According to Figure 3c, each VG consists of a delta-shaped pair of vanes with a uniform length, $L = 2h_{VG}$, and a certain VG height, $h_{VG}$, given in %$c$. Figure 3d shows that the distance between the two vanes is $d = 3h_{VG}$, facing each other in an angle $\beta \approx \pm 18°$. Furthermore, the spacing between the center points of two VGs is $D = 7h_{VG}$, see Figure 3b. The positions 0, 1 and 2 indicate the spanwise wake rake locations, as discussed in Sect. 4.2.3.

### 1.5 Combining Vortex generators and Gurney flaps

Despite the large body of literature on both PFC devices, the simultaneous use of GFs and VGs is less profoundly researched. Storms et al. (1994) investigated one such configuration in the NASA Ames Research Center based on the NACA4412 at $Re$ = 2·10⁶. Combining $h_{GF} = 1.25$ %$c$ with $h_{VG} = 0.5$ %$c$ at $x_{VG} = 12$ %$c$ and a spacing between the VGs of $D = 6h_{VG}$, stall is delayed by around 5° and $c_{l,max}$ is increased by 36 % improving $L/D$ at elevated AoA. However, at low and moderate AoA, the combined drag penalty leads to reduced $L/D$ ($\alpha$). Fuglsang et al. (2003) conducted extensive research based on the Riso-B1-24 in the VELUX wind tunnel of the DTU at $Re = 1.6·10⁶$. The Riso-B1 family is dedicated to rotor blades of multi MW wind turbines. The tested configurations consisted of $h_{GF} = 1$ %$c$ combined with $h_{VG} = 1$ %$c$ at $x_{VG} = 20$ %$c$ and $D = 4.2h_{VG}$ delaying separation by approximately 3°, coupled with an increase in $c_{l,max}$ of 34 %. Additionally, ZZ tape was applied on the suction side with $h_{ZZ} = 0.35$ mm at $x_{ZZ} = 5$ %$c$. Still, the combined effect of GFs and VGs delayed stall by approximately 3° and increased $c_{l,max}$ by 29 %. Fuglsang et al. (2004) concluded that, despite a small $L/D$ decrease, the combination of both devices *"(...) could provide an attractive choice for the root part of a wind turbine blade where reduction of solidity is a key issue to reduce blade costs."*. Besides, these results are in good agreement with comparable tests using the prior Riso-A1-24 airfoil design, as previously reported by Fuglsang et al. (1999). In a more recent study, Li-shu et al. (2013) performed experiments on the WA251A airfoil at the Northwestern Polytechnical University of Xi'an at $Re = 3·10⁶$. Applying $h_{GF} = 0.9$ %$c$ and 1.25 %$c$ together with $h_{VG} = 0.5$ %$c$ at $x_{VG} = 21$ %$c$, $c_{l,max}$ is increased by 18.6 % delaying stall by approximately 2°. The authors report *"remarkable improvements"* implementing both PFC devices simultaneously. In all mentioned reports of this section, the combined effect of GFs and VGs appears to be beneficial compared to the corresponding VG (only) configuration.

### 2. Airfoil simulations

In preparation for the wind tunnel tests, the simulation software XFOIL is used to determine the appropriate dimensions of each PFC device in relation to the local boundary layer thickness of the corresponding airfoil.



## 2.1 Airfoils

Figure 4a shows the airfoils that are tested throughout the wind tunnel experiments: the NACA63(3)618, the AH93W174 and
the DU97W300. They are applied at different sections of large rotor blades, see Figure 4b. The main airfoil specifications are
summarized in Table 1.

(a)

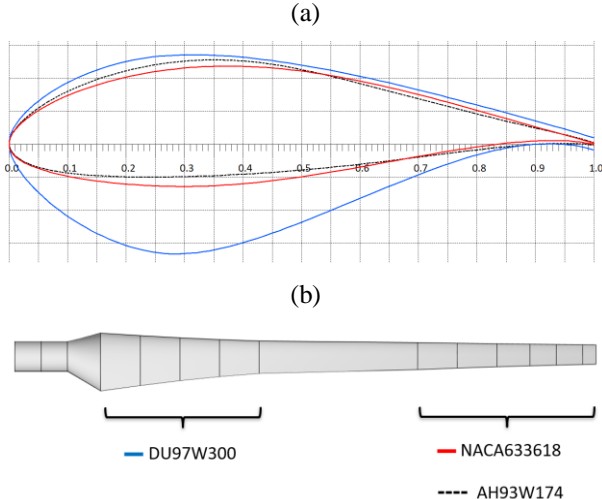

(b)

— DU97W300      — NACA633618

---- AH93W174

Figure 4. NACA63(3)618, AH93W174, DU97W300. (a) Airfoil coordinates normalized by the chord length. (b) Position of airfoils on a
generic rotor blade.

Table 1. Maximum thickness, maximum camber and trailing edge thickness. Chord position in brackets. All values in %c.

|  | $h_{th,max}$ $(x)$ | $h_{camber,max}$ $(x)$ | $h_{TE}$ |
|---|---|---|---|
| **NACA63(3)618** | 18.0 (34.0) | 3.0 (53.7) | 0.17 |
| **AH93W174** | 17.4 (33.0) | 4.0 (38.0) | 0.33 |
| **DU97W300** | 30.0 (29.3) | 2.1 (80.5) | 1.75 |


The NACA63(3)618 is part of the six-digit wing sections developed by the National Advisory Committee for Aeronautics
(NACA) for use on high speed aircrafts, see Abbott and von Doenhoff (1959). The NACA 63 and 64 families are still popular
for the design of large rotor blades, especially in the mid to tip region, see Timmer (2009). The NACA63(3)618 is characterized
by a relatively high lift and low drag performance due to the extended laminar BL. The main downside is the sensitivity to
roughness effects, as shown in Sect. 4.1.1. The AH93W174 is a wind turbine airfoil for the mid to tip blade region developed
by the Institut für Aerodynamik und Gasdynamik (IAG) at Stuttgart University, see Althaus (1996). Compared to the
NACA63(3)618, the maximum thickness is similar, but the maximum camber is higher, thus leading to a steeper lift slope.
Furthermore, $L/D_{max}$ is shifted towards more elevated AoA covering a relatively wide range of high performance. However,
the aerodynamic efficiency is degraded by the effect of forced LE transition, see Sect. A1. The DU97W300 is dedicated to the
root to mid blade region, developed by the Delft University (DU), see Timmer and van Rooij (2003). It is characterized by the





limited upper surface thickness and relatively low maximum camber, see Figure 4a. Therefore, the pressure recovery is milder avoiding premature separation on the suction side, while improving its sensitivity to roughness effects, see Sect. 4.2.1. The lift penalty is compensated for by the pronounced S- shape of the PS contour approaching the TE, as explained by van Rooij and Timmer (2003). In fact, this aft-loading tail changes the flow field in a similar way than the GF, as previously discussed in
Sect. 1.2.

## 2.2 XFOIL simulations

The 2D airfoil performance is simulated with the panel code XFOIL developed by Drela (1989). The freely available and widely recognized software is based on an viscid-inviscid interaction scheme, which was validated, amongst many others, by Timmer and Schaffarczyk (2004) and Fuglsang et al. (2004). Apart from the airfoil coordinates, including the finite TE
thickness, the software requires the chord-based Re number, here $Re = 1.5 \cdot 10^6$ and the AoA range, here $-5° < α < 20°$, as input parameters. The location of the free laminar-to-turbulent BL transition is modeled by means of the $e^N$ method. The amplification factor, or N criterion, describes the level of both the surface roughness and the inflow turbulence intensity. The default value, $N = 9$, refers to clean conditions, i.e. assuming a completely smooth surface and laminar inflow conditions that are found in low turbulence wind tunnels. In this study, $N = 5$ is chosen to account for the relatively strong turbulence intensity
of approximately 0.3%, see Sect. 3.1. In the so-called tripped case, the transition location is fixed at a static chordwise position, $x_{tr}$, on both the SuS and the PS.

The appropriate height of each PFC device is determined in relation to the local BL thickness, $δ$, which is defined as the normal distance between the solid surface and the first streamline reaching 99% of the axial free flow velocity. XFOIL calculates the
BL displacement thickness, $δ^*$, i.e. describing the distance by which the free flowing streamlines are displaced from the solid surface due to the existence of the BL. According to Schlichting and Gersten (2000), the laminar BL thickness on a flat plate at zero incidence is approximately three times the BL displacement thickness,

$$δ \approx 3δ^*. \qquad (1)$$

Eq. ( 1 ) is also valid for thin airfoil shapes. According to Baldacchino et al. (2018), the turbulent BL thickness is related to $δ^*$ and the momentum thickness $θ$,

$$δ \approx θ \left( 3.15 + \frac{1.72}{\left(\frac{δ^*}{θ}\right)-1} \right) + δ^*. \qquad (2)$$

## 2.3 Zig Zag tape

The baseline configurations include both the free and the forced BL transition. In the so-called tripped case, ZZ tape is applied alongside the complete airfoil span on both the SuS and the PS, as shown in Figure 3a. The ZZ tape height is selected in relation to the laminar BL thickness, see Eq. ( 1 ), at the corresponding chord positions, $x_{ZZ}$, and the design AoA, $α_{opt} = α$ ($L/D_{max}$), so that





$$h_{ZZ} \leq \delta(\alpha_{opt}, x_{ZZ}) \leq 3\delta^*(\alpha_{opt}, x_{ZZ}), \tag{3}$$

where $x_{ZZ,SuS} = 5\,\%c$ and $x_{ZZ,PS} = 10\,\%c$.

Table 2. XFOIL simulations of the boundary layer thickness according to Eq. ( 1 ).

|  | $\alpha_{opt}$ [°] | $\delta_{SuS}$ [mm] | $\delta_{PS}$ [mm] |
|---|---|---|---|
| **NACA63(3)618** | 5 | 0.51 | 0.55 |
| **AH93W174** | 7 | 0.53 | 0.53 |
| **DU97W300** | 9 | 0.54 | 0.42 |

According to Table 2 and applying Eq. ( 3 ), the NACA63(3)618 and the AH93W174 are both equipped with $h_{ZZ} = 0.4$ mm, and the DU97W300 with $h_{ZZ} = 0.3$ mm. The width of the turbulator tape is 12 mm and the angle between its serrations is 60°.

These characteristics are in close agreement with comparable wind tunnel tests at the DU and the DTU, see Sect. 4.

**2.4 Mini Gurney flaps**

Previously, the authors reviewed wind tunnel studies of 9 different DU and NACA airfoils at $1\cdot10^6 < Re < 2\cdot10^6$, as described in Alber et al. (2017). The lift and the drag increase between the GF and the baseline configurations, $\Delta c_l$ and $\Delta c_d$, were modelled with a surface best fit, resulting in the first degree polynomial functions ($R^2 > 0.9$),

$$\Delta c_l(\alpha) = c_{l,GF}(\alpha) - c_l(\alpha) = K_1 h_{GF} + K_2 c_l(\alpha) + K_3, \tag{4}$$

$$\Delta c_d(\alpha) = c_{d,GF}(\alpha) - c_d(\alpha) = K_4 h_{GF} + K_5 c_d(\alpha) + K_6, \tag{5}$$

where $K_i$ are the polynomial coefficients for $0.2\,\%c \leq h_{GF} \leq 2\,\%c$, for $\Delta c_l, \Delta c_d > 0$.

The model revealed a non-proportional dependency on the GF height: diminishing $h_{GF}$, $\frac{\Delta c_l(\alpha)}{h_{GF}}$ increased, whereas $\frac{\Delta c_d(\alpha)}{h_{GF}}$ decreased. Consequently, applying Eq. ( 4 ) and ( 5 ), the impact of the GF height on the aerodynamic efficiency was estimated by means of the following condition,

$$\frac{(c_l + \Delta c_l)(\alpha)}{(c_d + \Delta c_d)(\alpha)} \geq \frac{c_l(\alpha)}{c_d(\alpha)} \equiv \frac{\Delta c_l(\alpha)}{\Delta c_d(\alpha)} \geq \frac{c_l(\alpha)}{c_d(\alpha)}, \tag{6}$$

for $c_l, c_d > 0$.

Based on the data evaluation of Alber et al. (2017), Eq. ( 6 ) was only satisfied for relatively small GFs in the range of $0.2\,\%c \leq h_{GF} \leq 0.5\,\%c$. In other words, the GF needs to be submerged *deeply* into the local BL to avoid an over-proportional drag, in relation to the lift increase, as previously postulated by Giguère et al. (1995), Kentfield (1996) and Giguère et al. (1997). Hence, the crucial design parameter with regard to the aerodynamic efficiency is the ratio between the GF height and the BL displacement thickness at the TE,

$$\frac{h_{GF}}{\delta^*}(\alpha). \tag{7}$$

Following from Eq. ( 7 ), the definition of a MGF hereby refers to an effective height of between one and two times the local $\delta^*$ at the design AoA, so that





$$\delta^*(\alpha_{opt}) \leq h_{MGF} \leq 2\delta^*(\alpha_{opt}).$$ ( 8 )

For $h_{MGF} < \delta^*$, the impact of the MGF on the airfoil performance becomes insignificant. Combining Eq. ( 2 ) and Eq. ( 8 ), an appropriate MGF height is estimated to be in the order of one quarter of the turbulent BL thickness at the TE,

$$h_{MGF} \approx 0.25 \cdot \delta(\alpha_{opt}).$$ ( 9 )

Within the boundary condition given by Eq. ( 8 ) and Eq. ( 9 ), the impact of a MGF appears to be beneficial in terms of the
lift performance, the stall behavior and the aerodynamic efficiency, as further discussed in Sect. 4.

Table 3. XFOIL simulations of the boundary layer thickness according to Eq. ( 2 ).

|  | $\alpha_{opt}$ [°] | $\delta^*_{clean}$ [%c] | $\delta_{clean}$ [%c] | $\delta^*_{ZZ}$ [%c] | $\delta_{ZZ}$ [%c] |
|---|---|---|---|---|---|
| **NACA63(3)618** | 5 | 0.17 | 1.28 | 0.28 | 2.07 |
| **AH93W174** | 7 | 0.11 | 0.82 | 0.23 | 1.66 |
| **DU97W300** | 9 | 0.25 | 2.16 | 0.35 | 2.85 |

According to Table 3 and applying Eq. ( 8 ) and Eq. ( 9 ), the MGFs are in the range of 0.1 %c < $h_{MGF}$ < 0.7 %c. It is reiterated
that $\delta^*$ increases significantly in case of forced LE transition due to the early expansion of the turbulent BL. This observation is crucial when applying MGFs in order to compensate for the decambering effects of ZZ tape, as discussed in Sect. 4 and 5. In summary, $h_{MGF} = 0.25$%c, 0.5 %c and $h_{GF} = 1$ %c are implemented to cover for both the clean and the tripped baseline cases. It is noted that $h_{GF} = 1$ %c is not considered a MGF configuration as per Eq. ( 8 ). All tested GFs consist of standard and equilateral angle profiles made of brass with an ascending wall thickness of 0.3 mm, 0.5 mm and 0.6 mm, in relation to the
size of the GF.

## 2.5 Vortex generators

In general, the chord position of the VG array, $x_{VG}$, is located upstream the mean separation line, $x_{sep}(\alpha_{cl,max})$, in order to delay stall, and downstream of the BL transition location, $x_{tr}(\alpha_{opt})$, in order to minimize the drag penalty, so that $x_{tr}(\alpha_{opt}) < x_{VG} < x_{sep}(\alpha_{cl,max})$. Applying Eq. ( 2 ), the VG height is determined in relation to the turbulent BL thickness on
the SuS, i.e. at the initiation of stall,

$$h_{VG} \geq \delta(x_{VG}, \alpha_{cl,max}).$$ ( 10 )

Table 4. XFOIL simulations of the boundary layer thickness according to Eq. ( 2 ).

|  | $\alpha_{cl,max}$ [°] | $x_{VG}$ [%c] | $\delta_{clean}(x_{VG})$ [%c] | $\delta_{ZZ}(x_{VG})$ [%c] |
|---|---|---|---|---|
| **NACA63(3)618** | 12 | 50 | 1.55 | 1.62 |
| **AH93W174** | 11 | 30 | 0.64 | 0.66 |
| **DU97W300** | 12 | 30 | 0.58 | 0.72 |



Table 4 shows that, at stall, $\delta$ is similar in both the clean and the tripped cases. Based on Eq. ( 10 ), an uniform VG height of $h_{VG}$ = 1.1%c is selected, including the thickness of the base panel, see Figure 3b. In case of the DU97W300 and the AH93W174, $x_{VG}$ = 30%c resembles a standard VG array in the root to mid region of rotor blades. Regarding the NACA63(3)618, a tip VG configuration is investigated at $x_{VG}$ = 50%c, where $h_{VG} < \delta$ in order to reduce additional drag. Next to the effect of stall delay, the primary objective is to maintain $L/D$ ($\alpha$) on a high level in the mid to tip blade region. According to Lin (2002), the optimum height of sub-BL or low- profile VGs is in the range of 20%$\delta$ < $h_{VG}$ < 50%$\delta$.

### 2.6 Summary

Figure 5a displays both the height and the chordwise location of each PFC device that is investigated in this section. Figure 5b and Figure 5c depict the vorticity caused by either VGs or MGFs, according to previous research efforts at the HFI of the TU Berlin. The wake interaction of the flow control mechanisms and its effects on the lift and drag performance is tested as part of the wind tunnel campaign, as described in the following sections of this report.

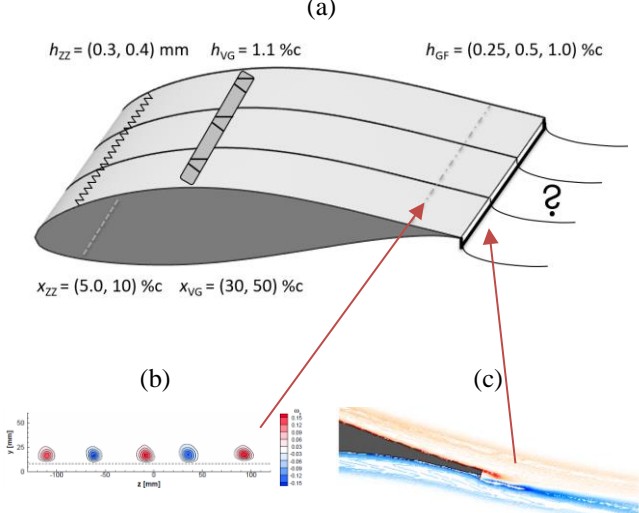

Figure 5. (a) Height and location of passive flow control devices. (b) PIV measurements of VG vortices on the NACA63(3)618 in spanwise view ($h_{VG}$ = 1.7%c at $x_{PIV}$ = 80%c and $Re$ = 1.3·10$^6$), reproduced from Mueller-Vahl et al. (2012). (c) CFD simulations of a Mini Gurney flap on the HQ17 in side view ($h_{GF}$ = 0.5%$c$ at $\alpha$ = 4.0° and $Re$ = 1·10$^6$), reproduced from Schatz et al. (2004a).

### 3. Experimental set-up

The wind tunnel test section, the measurement methods and the data reduction process are specified, including the force balance for the lift, and the wake rake for the drag measurements at a constant Reynolds number of $Re$ = 1.5·10$^6$.



### 3.1 Test section

The experiments are conducted in the large closed-loop wind tunnel of the HFI at the TU Berlin. The airfoil test section is 2 m in width and 1.44 m in height. It consists of a 2.5 m long removable structure, the so-called airfoil box, which is attached to the duct outlet, see Figure 6a. The contraction ratio is 6.25 : 1 and the complete length of the test section is 5 m. Between 1998 and 2000, the airfoil box was designed, constructed and integrated into the wind tunnel by Meyer (2000) in cooperation with the Deutsche Zentrum für Luft- und Raumfahrt e.V. (DLR). Since then, it has provided the experimental set-up for numerous research projects with a focus on passive and active flow control devices, such as Bechert et al. (2000), Meyer et al. (2006), Pechlivanoglou et al. (2011), Mueller-Vahl et al. (2012) and Bach et a. (2014).

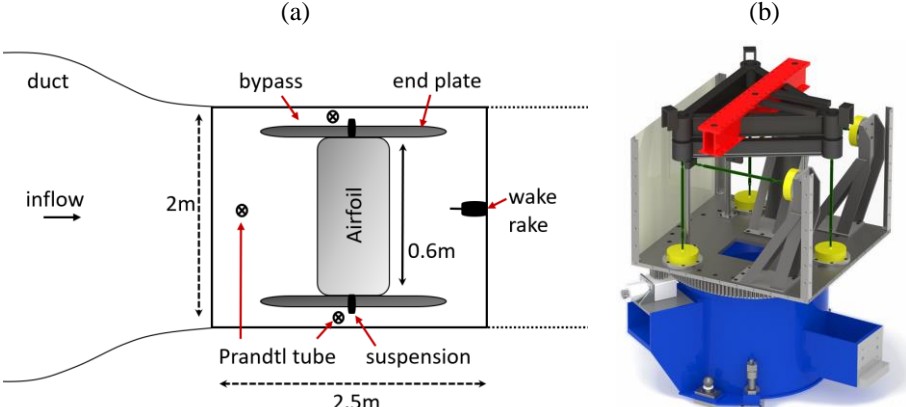

Figure 6. (a) Airfoil test section in top view. (b) Force balance underneath the test section in side view, load cells in yellow.

The inflow velocity is determined from the static pressure difference between the inlet and the outlet of the duct. For additional monitoring purposes, a Prandtl tube is mounted approximately $x = c = 100\%c$ upstream the airfoil LE. The airfoil model, or wing, is positioned in the centre of the box, as displayed in Figure 6a. It is enclosed by two smooth end plates, or splitter walls, that are parallel to the tunnel in order to minimize the influence of the wall BL. The velocities inside the 0.25 m wide bypass channels are measured via two separate Prandtl tubes to obtain the effective inflow or reference velocity. The suspension is decoupled from the end plates and the tunnel walls. Hence, the wing is directly mounted to the platform of the permanently installed force balance underneath the test section, see Figure 6b. The AoA is controlled by means of a stepping motor with an accuracy of 0.1°, which is part of the suspension. The airfoil models include the NACA63(3)618, the AH93W174 and the DU97W300, see Figure 4. All three wings were CNC milled from a solid block of Obomodulan^TM, as described by Pechlivanoglou (2013). The chord length is 0.6 m and the span is 1.54 m resulting in an aspect ratio of 2.56.





## 3.2 Measurement methods

### 3.2.1 Force balance

The lift, drag and side forces of the airfoil model are directly transferred to the six component force balance. The load cells consists of strain gauges generating voltage signals that are proportional to the incoming forces. Each signal is digitalized by a CompactDAQ System of National Instruments with a sampling rate of 5 kHz. The data is recorded using a LabVIEW user interface, including forces, velocities and environmental conditions, i.e. air density and kinematic viscosity, all of which are

automatically converted to average results. The LabVIEW interface also contains the AoA control in terms of range, steps and measurement duration. According to Meyer et al. (2016), the uncertainty of the averaged $c_l$ and $c_d$ measurements is 0.2%. The $c_l$ ($\alpha$) results show good agreement with literature data, as discussed in Sect. 4. However, since its implementation in the wind tunnel, the current set-up has been characterized by elevated $c_d$ ($\alpha$) results. The reasons are the small gaps between the wing and the end plates leading to suction effects. Moreover, a turbulent BL is formed on the end plates triggering separation on the

outer parts of airfoil model, i.e. in the vicinity of splitter walls. Both effects are 3D and therefore detected in the form of increased drag values, see Meyer (2000) and Meyer et al. (2016). For the purpose of this study, a wake rake is designed, constructed and implemented into the test section aiming at 2D drag measurements.

### 3.2.2 Wake rake

The wake rake method is widely recognized to determine 2D drag coefficients at pre-stall conditions, see Barlow et al. (1999)

and the references given throughout Sect 4.

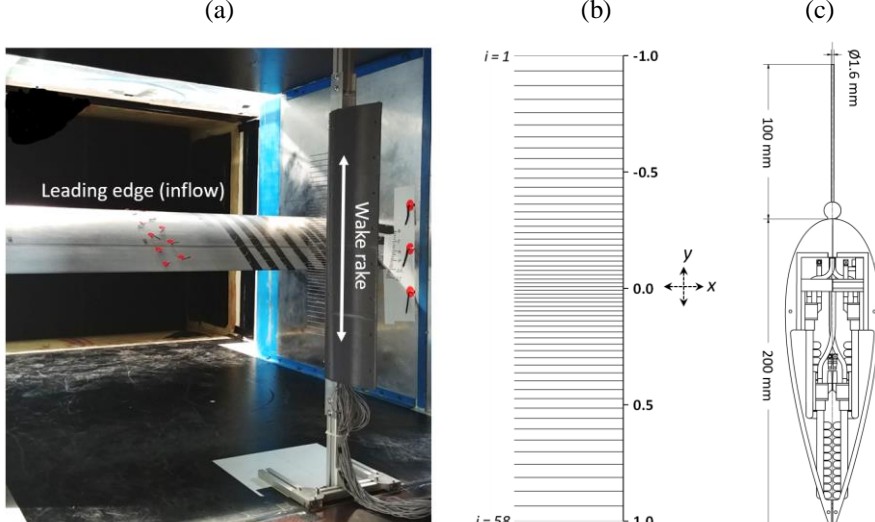

Figure 7. (a) Test section during wind tunnel measurements including airfoil model, one of the end plates and wake rake. (b) Normalized vertical rake tube positions, $y$, and numbering, $i$, in side view. (c) Cross section of wake rake casing and pressure tube in top view.





Figure 7a shows the present wake rake, which is positioned downstream the airfoil trailing edge at $x = c = 0.6$ m. This distance is necessary for the flow to return to the static pressure level in the wind tunnel, see Barlow et al. (1999). Figure 7b illustrates

that the rake consists of a straight vertical line of 58 pitot tubes, each measuring the total pressure, $p_{total}$. The normalized vertical positions are defined as $y = 0.0$ for the center, $y_{i=1} = -1.0 = -250$ mm for the uppermost tube and $y_{i=58} = 1.0 = 250$ mm for the lowest tube. The total rake span is 0.5 m. The spacing between the tubes is smallest towards the center with $\Delta y_{min} = 4$ mm, and widest towards top and bottom with $\Delta y_{max} = 16.5$ mm, i.e. the data resolution is highest towards the center. The 3D printed casing consists of the symmetrical NACA0030 profile, see Figure 7c. The distance between the LE and the orifice of each

pitot tube is 100 mm, where the impact of the rake on the flow is considered negligible. The static pressure, $p_{static}$, is determined as the average value of two Prandtl tubes that are installed inside the downstream plane of the rake, one on top and one below the casing. The differential pressure of each tube, $\Delta p(y_i)$, is measured with single HDOM010 sensors by First Sensor, which are installed inside the casing. They are connected with flexible silicon tubes, each shorter than 200 mm, in order to avoid dynamic feedback effects. The accuracy of each sensor is given with 0.1% of the full scale range of 1000 Pa under nominal

conditions. At the constant inflow velocity of $u_{inf} = 40$ m/s, or 900 Pa $< \Delta p <$ 1000 Pa, the sensor accuracy is in the order of 1 Pa and therefore considered to be minor. The voltage signal is digitalized by a separate CompactDAQ system at a sampling rate of 10 kHz and recorded by a Labview user interface, both of which are decoupled from the DAQ system of the force balance, see Sect. 3.2.1. All sensors are calibrated by means of a mobile pressure calibration devices. Preparatory differential pressure tests in the empty test section show very good agreement between all rake sensors and the inflow Prandtl tube. After

installing each airfoil model, the vertical center of the rake, $y = 0.0$, is roughly aligned with the maximum pressure loss at the corresponding design AoA. In this way, the static rake span covers the complete AoA range, $-5° < \alpha < 17°$, hence avoiding the installation of a vertical traversing system, see Figure 18, The raw data is post-processed using a specific Matlab script. The wake pressure loss and the drag coefficient is determined at each static AoA following Eq. ( 11 ) to ( 15 ).

1. The time-resolved total as well as static pressure is averaged in order to determine the differential pressure,

$$\Delta \bar{p}_i = \bar{p}_{total}(y_i) - \bar{p}_{static}, \tag{11}$$

where ρ refers to the air density in kg/m³ and $i = 1,2…58$ to each rake tube.

2. The reference pressure of the free flow is calculated as the mean value of the two uppermost and the two lowest rake tubes, see Figure 7b, so that,

$$\Delta \bar{p}_{ref} = 0.25 \cdot (\Delta \bar{p}_1 + \Delta \bar{p}_2 + \Delta \bar{p}_{57} + \Delta \bar{p}_{58}). \tag{12}$$

The pressure coefficient, $c_{pi}$, is defined as,

$$c_{pi} = \frac{\Delta \bar{p}_i}{\Delta \bar{p}_{ref}}, \tag{13}$$

describing the characteristic pressure loss at each AoA, see Figure 18.

3. The uncorrected total drag contribution of each rake tube, $c_{di}$, becomes,

$$c_{di} = \sqrt{c_{pi}} - c_{pi}. \tag{14}$$





4. The uncorrected total drag coefficient, $c_{d,raw}(\alpha)$, is numerically integrated over the spacing between the rake tubes using the trapezoid rule, so that,

$$c_{d,raw}(\alpha) = \frac{1}{c}\sum_1^{58}(c_{di} + c_{di+1}) \cdot (y_{i+1} - y_i),\qquad(15)$$

where $c$ is the airfoil chord length and $y_i$ the normalized position of each rake tube, as illustrated in Figure 7b.

### 3.2.3 Wind tunnel correction

The 2D airfoil performance is modified by the wind tunnel walls compared to equivalent free flow conditions. The reasons for that are: 1. the solid blockage effect leading to the constriction of the curved streamlines around the airfoil model and 2. the wake blockage effect leading to the constriction of the curved streamlines in the wake. Following Barlow et al. (1999), the total blockage, ε, is the sum of the solid and the wake blockage factors,

$$\varepsilon = \varepsilon_{solid} + \varepsilon_{wake} = \Lambda\frac{\pi^2}{48}\left(\frac{c}{h_{wt}}\right)^2 + \frac{c}{4h_{wt}}c_{d,raw} = \Lambda\mu + \frac{c}{4h_{wt}}c_{d,raw},\qquad(16)$$

for ε > 0,

where Λ refers the so-called body shape factor, which is related to the maximum airfoil thickness, $h_{wt} = 1.44\ m$, which is the height of the test section. Moreover, $\mu = \frac{\pi^2}{48}\left(\frac{c}{h_{wt}}\right)^2$ is introduced as an auxiliary constant.

Based on Eq. ( 16 ), the blockage correction is applied on the following parameters at each static AoA,

$$c_d = c_{d,raw}(1 - 3\varepsilon_{solid} - 2\varepsilon_{wake}),\qquad(17)$$

$$c_l = c_{l,raw}(1 - \mu - 2\varepsilon),\qquad(18)$$

$$Re = Re_{raw}(1 + \varepsilon),\qquad(19)$$

$$\alpha = \alpha_{raw} + \frac{57.3\mu}{2\pi}(c_{l,raw} + 4c_{m,raw}),\qquad(20)$$

$$c_m = c_{m,raw}(1 - 2\varepsilon) + 4\mu c_l,\qquad(21)$$

where $c_m$ refers to the moment coefficient at 0.25c.

Eq. ( 17 ) to ( 20 ) are directly implemented into the data post-processing Matlab script.

### 3.3 Test matrix

The inflow velocity, $u_{inf} = 40$ m/s, corresponds to a constant Re number of approximately $1.5 \cdot 10^6$. The free stream turbulence intensity of the empty wind tunnel is estimated with the inflow Prandtl tube and amounts to less than 0.3 %. The AoA ranges from -5° < α < 17°, in steps of 1°. At each static AoA, there is a buffer of 4 s for the flow to settle, after which data is recorded for another 5 s. Hence, the total number of samples is $n = 5 \cdot 10^4$ for each rake sensor and $n = 2.5 \cdot 10^4$ for each load cell of the

force balance. Before each test run, all sensors are subjected to a zero-offset measurement at standstill in order to reduce experimental errors.

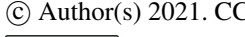



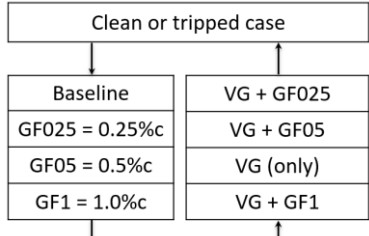

Figure 8. Test matrix and designation of all tested configurations.

Figure 8 summarizes the sequence of measurements for each airfoil starting with the clean baseline and followed by the three
GF (only) configurations. Next, the VG array is installed and tested in combination with GF1. Subsequently, GF1 is removed
in order to test VG (only), followed by the remaining configurations, VG + GF05 and VG + VG025. In the next round, ZZ
tape is attached and the test matrix is repeated. Each complete cycle, clean or tripped, is measured during the same day, i.e.
within less than 24 h, for the environmental conditions to remain as constant as possible.

## 4. Experimental results

The wind tunnel results are presented separately for the NACA63(3)618 (mid to tip blade region) and the DU97W300 (root to
mid blade region). For clarity, the results of the AH93W174 are included as Appendix A. The baseline measurements consist
of the clean and the tripped cases. They are compared to literature data in order to validate the experimental set-up. Next,
different combinations of MGFs and VGs are assessed by means of characteristic parameters, i.e. the lift performance, the stall
behavior and the aerodynamic efficiency.

### 4.1 NACA63(3)618

#### 4.1.1 Baseline

Figure 9 shows the clean and the tripped polar curves. The drag results are valid until stall at $\alpha_{cl,max} = 10.5°$ and the lift curves
are measured until the post-stall AoA of 18.5°, see Figure 9a. As expected, ZZ tape with $h_{ZZ} = 0.4$ mm manifests itself in a lift
decrease, coupled with a significant drag increase. The design point decreases from $\alpha_{opt,clean} = 6.4°$ to $\alpha_{opt,ZZ} = 5.4°$ and the
corresponding aerodynamic efficiency drops from $L/D_{max,clean} = 109$ to $L/D_{max,ZZ} = 60$, see Figure 9b. For clarity, characteristic
lift and L/D values are summarized in Table 5.



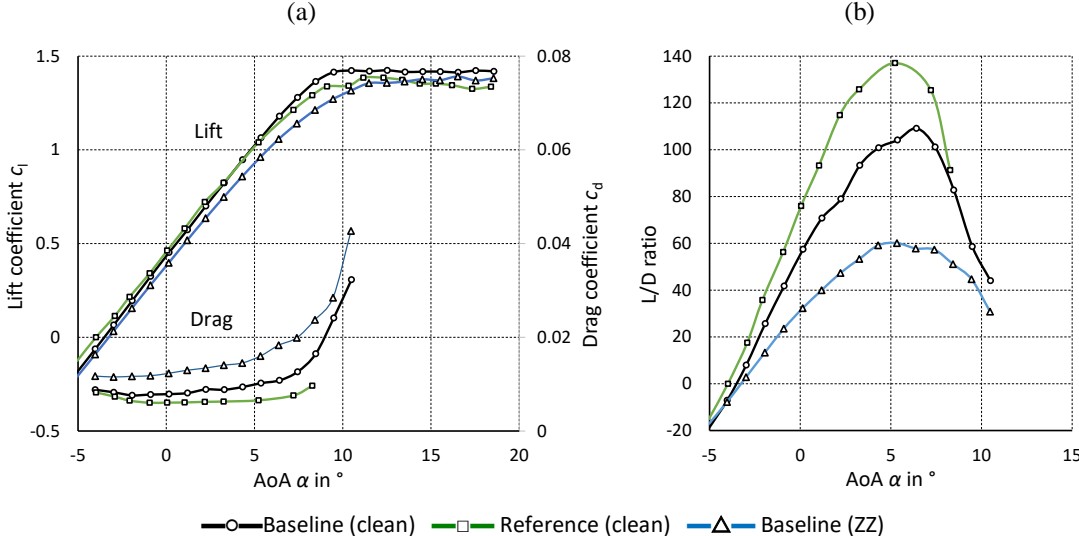

Figure 9. NACA63(3)618. Clean and tripped baseline cases at Re = 1.5·10$^6$ compared to reference data from Abbott and von Doenhoff (1959) at Re = 3·10$^6$. (a) Lift and drag coefficients. (b) L/D ratio.

The clean baseline measurements are compared to literature data from the airfoil catalogue of Abbott and von Doenhoff (1959). The NACA63(3)618 was tested at the closed-throat NASA Langley low turbulence pressure wind tunnel. The inflow turbulence was below 0.005 % at $Re = 3.0·10^6$. Lift was determined by means of the reaction pressure difference between wind tunnel floor and ceiling and drag with a wake rake, both including wall and blockage corrections. Figure 9a shows good agreement between the lift curves. The separation points, $\alpha_{cl,max}$, and the design points, $\alpha_{opt}$, are very similar, even though $c_{l,max}$

is slightly more elevated compared to the reference, which is probably due to the different measurement methods. The smaller drag and thus higher $L/D$ results are explained by the fact that, in the reference case, double the Reynolds number is applied, see Figure 9b. Besides, the inflow turbulence is significantly stronger in case of the current measurements leading to earlier transition at elevated AoA and thus higher drag values.

**4.1.2 Mini Gurney flaps**

Timmer and van Rooij (2003) and Fuglsang et al. (2003) report no significant changes comparing the effect of rectangular and triangular GFs of identical height, apart from minor differences in drag. In order to verify this observation, the NACA63(3)618 is equipped with both GF profiles, as previously illustrated in Figure 1a: thin angle sections made of brass versus isosceles triangles made of a standard thermoplastic material. Figure 10 compares the lift over drag in both the clean and the tripped cases looking at each GF configuration separately. It is noted that the smallest triangular size, $h_{MGF,\Delta} = 0.33$ %c is larger than

the corresponding rectangular profile, $h_{MGF,L} = 0.25$ %c, see Figure 10a. Overall, all wedge shaped profiles shows a slight decrease in both lift and drag, which is visible in terms of GF05 and GF1, see Figure 10b and c. Nonetheless, the agreement between the two profiles is considered satisfying. In the remainder of this report, all GF configurations refer to the rectangular, or L-shaped, profile.



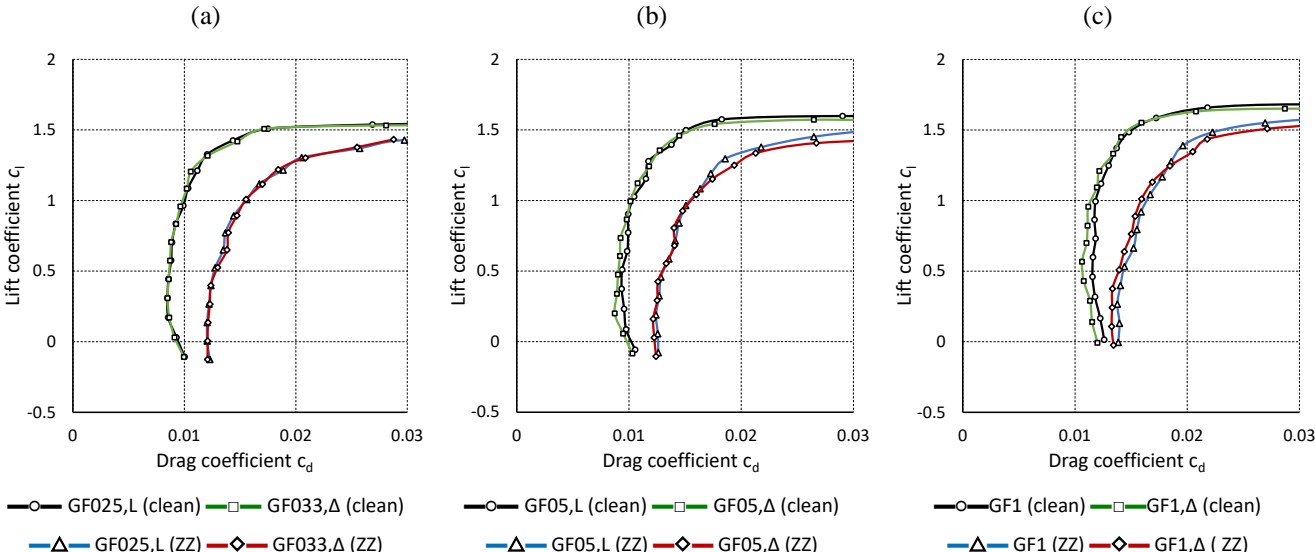

Figure 10. NACA63(3)618. Rectangular (L) versus triangular (Δ) Gurney flap profiles. Lift over drag curves in clean and tripped cases (a)
$h_{MGF,L} = 0.25$ %c and $h_{MGF,\Delta} = 0.33$ %c. (b) $h_{MGF} = 0.5$ %c. (c) $h_{GF} = 1$ %c.

Next, Figure 12a displays the pressure deficit or momentum loss in the wake of the GF configurations based on Eq. ( 13 ),
which is then converted into the drag coefficients. In the clean case, at $\alpha_{opt} = 6.4°$, the $c_{pi}$ curves correspond to attached flow.
As expected, the pressure deficit and thus drag is increased in relation to the GF height. Moreover, the position of the horizontal
centerline of the wake dent is characteristic for the downwash angle, which is proportional to lift, as discussed in Sect. 1.2.
Hence, the minimum of the $c_{pi}$ curves descend towards the wind tunnel floor for increased GF heights. In the tripped case, the
wake pressure deficit is more pronounced due to higher drag, and the downwash angle is smoother due to lower lift values,
see Figure 12b.

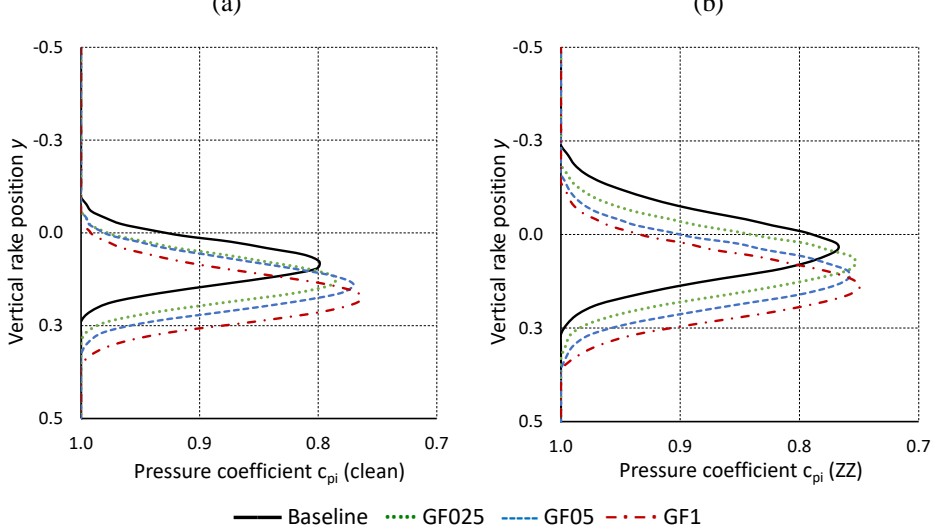



Figure 11. NACA63(3)618 at $\alpha = 6.4°$. Gurney flaps. Pressure coefficients over vertical wake rake positions. (a) Clean case. (b) Tripped case.

Furthermore, the fluctuations of the wake pressure measurements are determined via the standard deviation of the raw data,

$$\sigma_{pi} = \sqrt{\frac{1}{n-1}\sum_1^n |\Delta p_i - \Delta \bar{p}_i|^2}, \qquad (22)$$

where $n = 5 \cdot 10^4$ is the number of samples of each pressure sensor and $\Delta \bar{p}_i$ refers to the average differential pressure at each AoA, see Eq. ( 11 ).

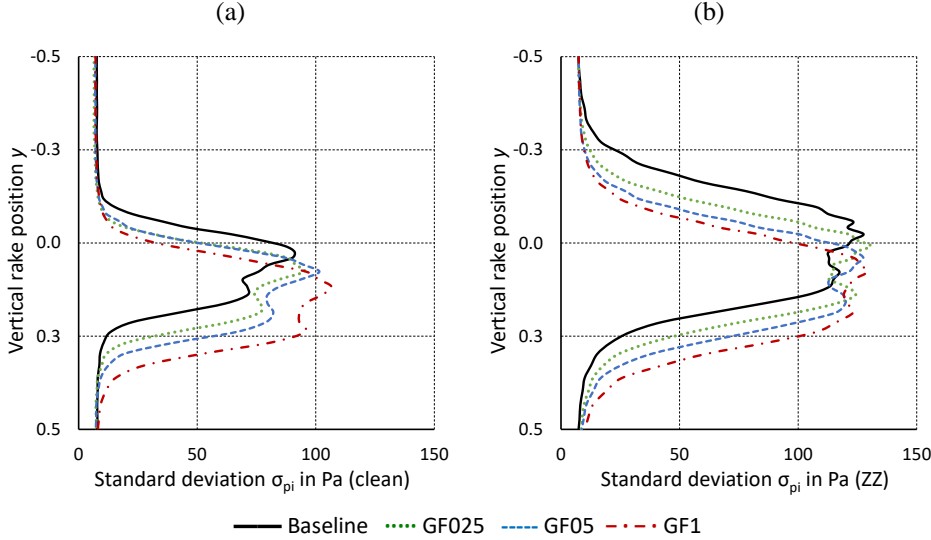

Figure 12. NACA63(3)618 at $\alpha = 6.4°$. Gurney flaps. Standard deviation of raw pressure data over vertical wake rake positions. (a) Clean case. (b) Tripped case.

Figure 12a shows that, in the clean case, the intensity of the wake unsteadiness is dependent on the GF height. Despite the offset due to the higher downwash angle, the $\sigma_{pi}$ curves are very similar between the clean baseline and the MGF configurations, as predicted by Bechert et al. (2000) and Schatz et al. (2004b), see Sect. 1.2. Moreover, Figure 12b shows that $\sigma_{pi}$ is much more pronounced in the tripped baseline case because of the thicker and more turbulent BL, so that the contribution of all GF configurations to $\sigma_{pi}$ appears to be marginal.


Next, Figure 13 illustrates the polar curves of all GF configurations referring to the NACA63(3)618. Depending on the GF height, both lift and drag is increased. Compared to the respective baseline, the shape of the polar curves and the stall behaviour is maintained, so that $\alpha_{opt,clean} = 6.4°$, $\alpha_{opt,ZZ} = 5.4°$ and $\alpha_{cl,max} = 10.5°$ in both the clean and the tripped cases. For clarity, characteristic lift and $L/D$ values are summarized in Table 5.






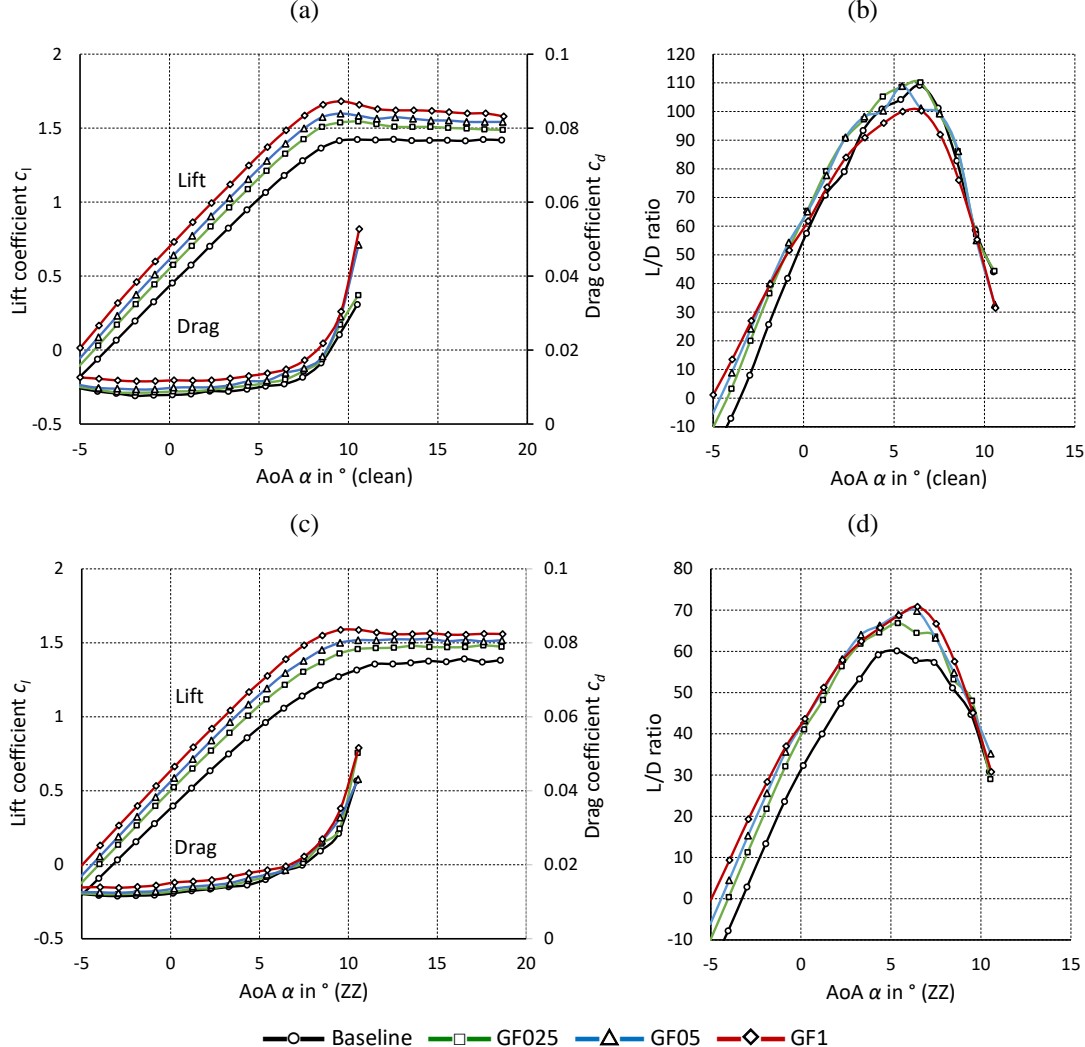

Figure 13. NACA63(3)618. Gurney flaps. (a) Lift and drag coefficients (clean). (b) L/D ratio (clean). (c) Lift and drag coefficients (ZZ). (d) L/D ratio (ZZ).

Table 5. NACA63(3)618. Gurney flaps. Characteristic values.

| | Clean | | Tripped | |
|---|---|---|---|---|
| | $c_{l,max}$ (10.5°) | $L/D_{max}$ (6.4°) | $c_{l,max}$ (10.5°) | $L/D_{max}$ (5.4°) |
| **Baseline** | 1.42 | 109 | 1.32 | 60 |
| **GF025** | 1.54 | 110 | 1.46 | 67 |
| **GF05** | 1.58 | 101 | 1.52 | 69 |
| **GF1** | 1.66 | 100 | 1.59 | 69 |

Looking at the clean cases, Figure 13b illustrates that $L/D_{max,clean}$ is slightly improved by GF025 and GF05. As such, GF025 provides the preferred results, while GF1 leads to an overall $L/D$ (α) decrease. In the tripped cases, the aerodynamic efficiency



is improved by all GF heights, see Figure 13d and Table 5. As discussed in Sect. 2.4, the local BL expands significantly due to forced LE transition, so that larger GFs appear to be more beneficial.

### 4.1.3 Vortex generators plus Mini Gurney flaps

Figure 14 illustrates the polar curves of all VG + GF configurations. Looking at the VG (only) cases, stall is delayed by around

2°, coupled with a substantial increase in $c_{l,max}$, see Figure 14a. Furthermore, VGs lead to a more abrupt stall behaviour and thus adverse load excursions, as previously reported by Mueller-Vahl et al. (2012). In all combined cases, the GF is added to the VG effect as a constant lift and drag increase. For clarity, characteristic lift and $L/D$ values are summarized in Table 6.

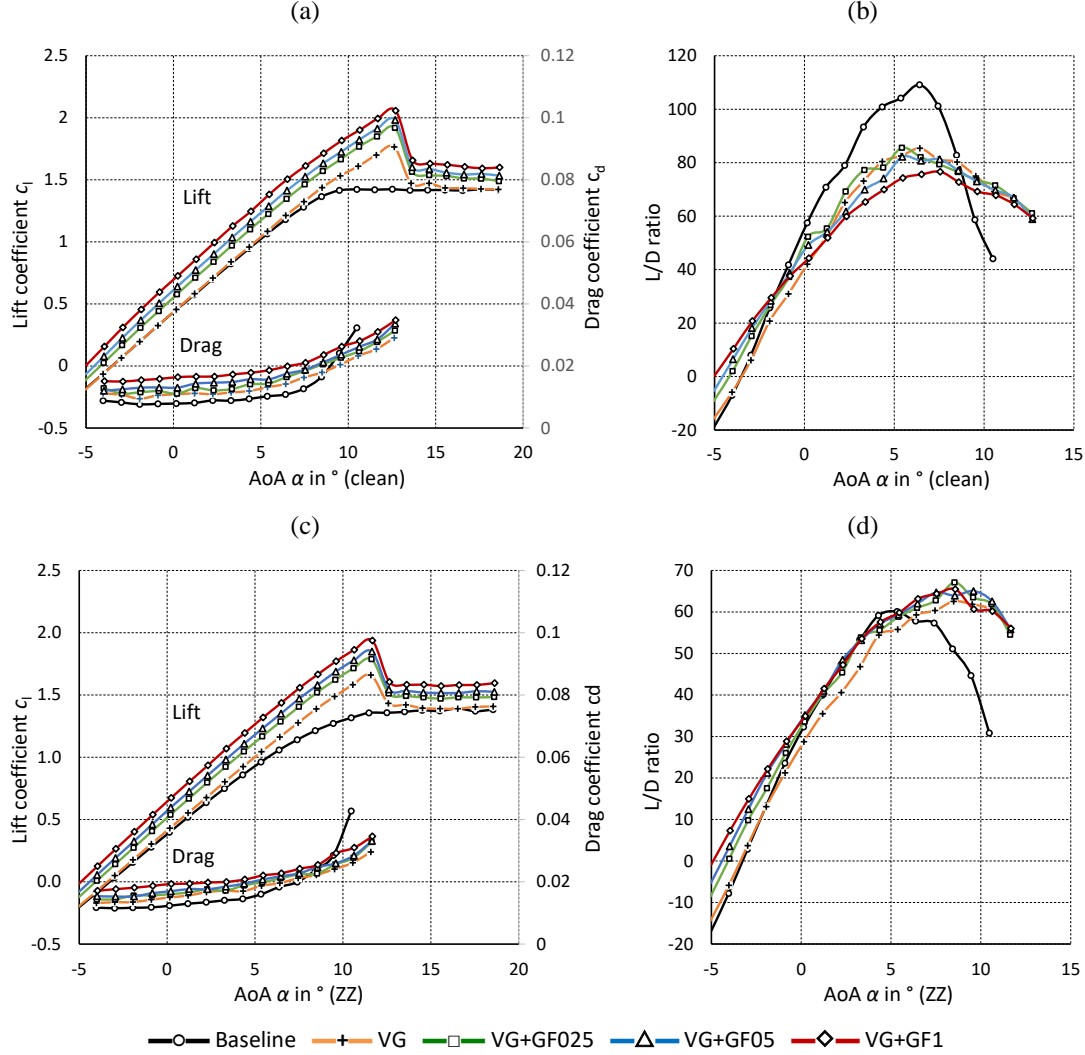

Figure 14. NACA63(3)618. Vortex generators and Gurney flaps. (a) Lift and drag coefficients (clean). (b) L/D ratio (clean). (c) Lift and drag coefficients (ZZ). (d) L/D ratio (ZZ).





Table 6. NACA63(3)618. Vortex generators plus Gurney flaps. Characteristic values.

| | Clean | | Tripped | |
|---|---|---|---|---|
| | $c_{l,max}$ (α) | $L/D_{max}$ (α) | $c_{l,max}$ (α) | $L/D_{max}$ (α) |
| **Baseline** | 1.42 (10.5°) | 109 (6.4°) | 1.32 (10.5°) | 60 (5.4°) |
| **VG** | 1.76 (12.7°) | 85 (6.4°) | 1.66 (11.6°) | 63 (8.5°) |
| **VG+GF025** | 1.92 (12.7°) | 82 (6.4°) | 1.79 (11.6°) | 67 (8.5°) |
| **VG+GF05** | 1.98 (12.7°) | 81 (6.4°) | 1.85 (11.6°) | 64 (8.5°) |
| **VG+GF1** | 2.06 (12.7°) | 76 (6.4°) | 1.94 (11.6°) | 65 (8.5°) |

According to Figure 14a and c, drag is reduced at elevated AoA due to the effect of stall delay. However, in the clean case, the drag penalty at low and moderate AoA causes $L/D_{max,clean}$ to decrease. Hence, the preferred results are achieved for VG + GF025 compared to the remaining configurations, see Figure 14b. Under tripped conditions, all PFC devices are capable of
maintaining $L/D$ at low and moderate AoA. Moreover, $L/D_{max,ZZ}$ increases significantly, as it is shifted by around 2° and thus recovering a wide area of otherwise separated flow in the range of 6.5° < α < 11.6°, see Figure 14d. Again, VG + GF025 is the preferred configuration. Overall, the polar graphs indicate that the VG is superposed by the GF effect. In fact, looking at both the lift and the $L/D$ curves of Figure 14, the combined cases prove to be beneficial compared to the VG (only) configurations. The reason for this phenomenon is suspected to be the beneficial wake flow interaction between both PFC devices, as
previously illustrated in Figure 5 and further discussed in the following section.

### 4.2 DU97W300

#### 4.2.1 Baseline

Figure 15 illustrates the clean and the tripped polar curves. In the tripped case, the stall angle decreases from $α_{cl,max,clean} = 12.6°$ to $α_{cl,max,ZZ} = 10.4°$ and the design AoA from $α_{opt,clean} = 9.5°$ to $α_{opt,ZZ} = 7.4°$, see Figure 15a. Hence, using $h_{ZZ} = 0.3$mm,
separation is initiated early, in fact only 1° below $α_{opt,clean}$. As a result, the aerodynamic efficiency drops from $L/D_{max,clean} = 88$ to $L/D_{max,ZZ} = 41$, see Figure 15b. For clarity, characteristic lift and $L/D$ values are summarized in Table 7.

The baseline measurements are compared to reference data of the DU, see Timmer and van Rooij (2003). The DU97W300 was tested in the closed-loop low turbulence wind tunnel. The free stream turbulence is 0.07 % at $Re = 3 \cdot 10^6$. Lift was
determined from the measured surface pressure distribution and drag by means of a wake rake positioned 60 %c downstream the airfoil TE, including wall and blockage corrections. Forced LE transition was triggered by $h_{ZZ} = 0.35$mm at $x_{SuS} = 5.0$ %c and $h_{ZZ} = 0.25$mm at $x_{PS} = 20.0$ %c, i.e. very similar to the current configuration.

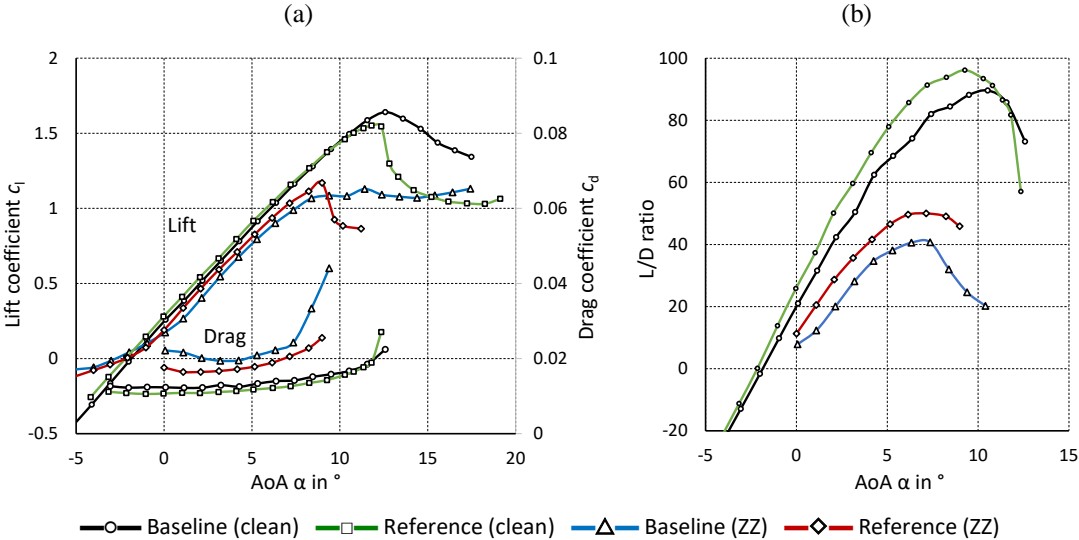

Figure 15. DU97W300. Clean and tripped baseline cases at $Re = 1.5\cdot10^6$ compared to reference data from Timmer and van Rooij (2003) at
$Re = 3\cdot10^6$. (a) Lift and drag coefficients. (b) Lift to drag ratio.

Figure 15a shows good agreement between the lift curves, including $\alpha_{cl,max}$. In both cases, the reference curves are characterized by a more abrupt stall behaviour compared to the current measurements. The smaller drag and thus larger $L/D$ values are explained by the higher Re number of the reference measurements, see Figure 15b. It is noted that both drag curves show abrupt PS separation at negative AoA due to the ZZ tape, which are not displayed in Figure 15a. In a related publication, van

Rooij and Timmer (2003) report the following specific values of the clean DU97W300 baseline at $Re = 2\cdot10^6$ rather than $3\cdot10^6$: $c_{l,max} = 1.55$ and $L/D_{max} = 91.7$. This is in excellent agreement with the current measurements at $Re = 1.5\cdot10^6$: $c_{l,max} = 1.64$ and $L/D_{max} = 89.5$.

### 4.2.2 Mini Gurney flaps

Figure 16 shows the polar curves of all GF configurations, which reinforce the observations presented in Sect. 4.1.2: 1. the

shape of the polar curves is basically maintained, 2. the lift and drag increase depends on the GF height, and 3. the beneficial GF effect on $L/D$ ($\alpha$) is more pronounced in the tripped case. For clarity, characteristic lift and L/D values are summarized in Table 7.

Figure 16b illustrates that, in the clean case, the aerodynamic efficiency is kept on a similar level applying either of the MGFs,

whereas $L/D_{max,clean}$ is decreased using GF1. The performance deterioration due to forced LE transition is alleviated by all GFs, with GF05 achieving the preferred results in terms of $L/D_{ZZ}$ ($\alpha$), see Figure 16d and Table 7.



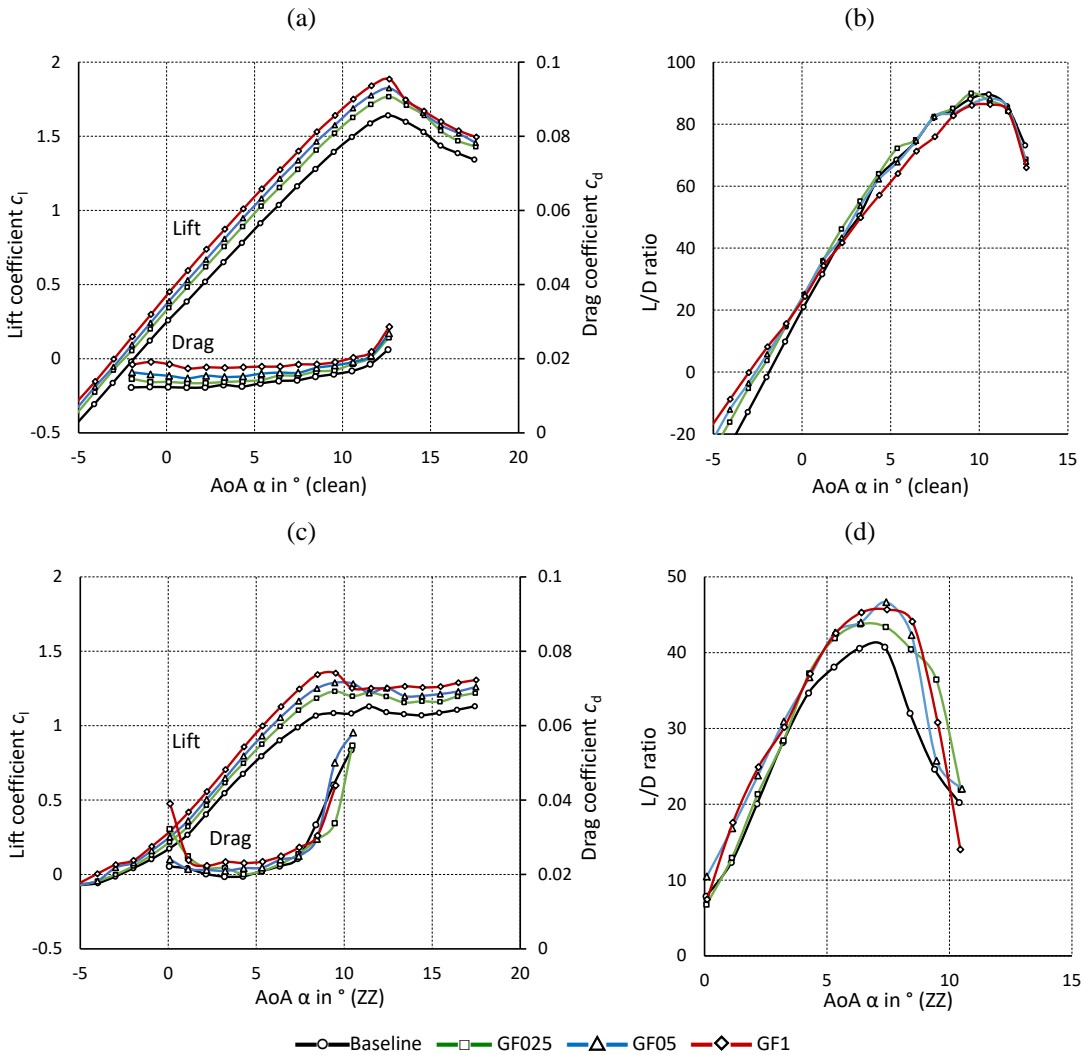

Figure 16. DU97W300. Gurney flaps. (a) Lift and drag coefficients (clean). (b) L/D ratio (clean). (c) Lift and drag coefficients (ZZ). (d) L/D ratio (ZZ).

Table 7. DU97W300. Gurney flaps. Characteristic values.

| | Clean | | Tripped | |
|---|---|---|---|---|
| | $c_{l,max}$ (12.6°) | $L/D_{max}$ (9.5°) | $c_{l,max}$ (10.4°) | $L/D_{max}$ (7.4°) |
| Baseline | 1.64 | 88 | 1.08 | 41 |
| GF025 | 1.77 | 90 | 1.20 | 43 |
| GF05 | 1.82 | 87 | 1.28 | 47 |
| GF1 | 1.89 | 86 | 1.25 | 46 |



### 4.2.3 Vortex generators plus Mini Gurney flaps

The flow field in the wake of GFs is considered 2D at pre-stall operation, see Sect. 1.2. As opposed to that, VGs induce counter rotating vortices into the BL that are released as scattered 3D structures into the TE wake, see Figure 5b. Hence, the VG effect is registered in the form of spanwise oscillations of the drag measurements by the wake rake, as reported by Baldacchino et. al (2018). The amplitude of these oscillations depend on both the size and the position of the VG array, the distance between the wake rake and the airfoil TE as well as the operational conditions, i.e. Re, AoA and inflow turbulence intensity. Figure 17 displays the spanwise dependency of the current drag measurements. In accordance with Figure 3b, position 0 stands for 0D, i.e. in the airfoil center and right in-between the VG vanes. At position 1, the rake is shifted to $0.25D = 11.4$ mm in the spanwise direction and position 2 refers to $0.5D = 22.8$ mm, i.e. right in between two adjacent VGs.

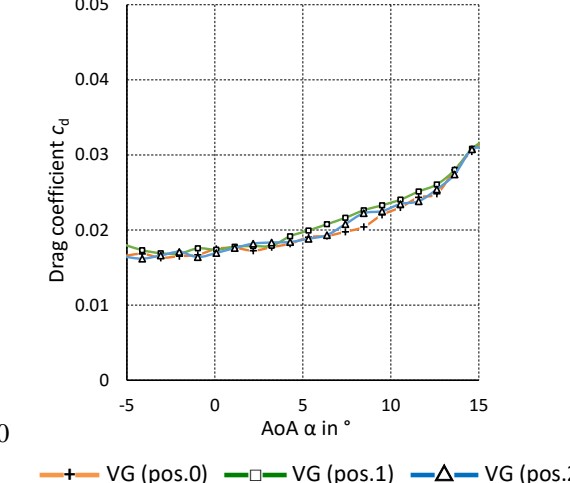

Figure 17. DU97W300. Clean VG (only) configurations. Drag measurements for different spanwise wake rake positions.

Figure 17 shows that the influence of the spanwise rake positions on the drag results is marginal at the relatively large distance of 100 %c between airfoil TE and rake tubes. Hence, all drag results that are presented in this report refer to the center position of the rake, i.e. the spanwise position 0.

Next, Figure 18 shows the pressure deficit or momentum loss in the wake of all clean VG + GF configurations. Figure 18a captures the pressure distribution at the design AoA, $\alpha_{opt,clean} = 9.5°$. All configurations are characterized by attached flow, so that the wake deficit and thus the drag coefficients are relatively small. The pre-stall lift increase due to the GFs is directly related to the downwash angle, as discussed in Sect. 4.1.2. Figure 18b illustrates the situation at $\alpha_{cl,max} = 12.6°$, where the baseline curve indicates that separation is initiated. The wake deficit extends towards the upper side of the rake due to the formation of the TE separation bubble on the airfoil SuS. At the same time, the VG + GF configurations delay separation leading to a much smaller wake deficit compared to the baseline.



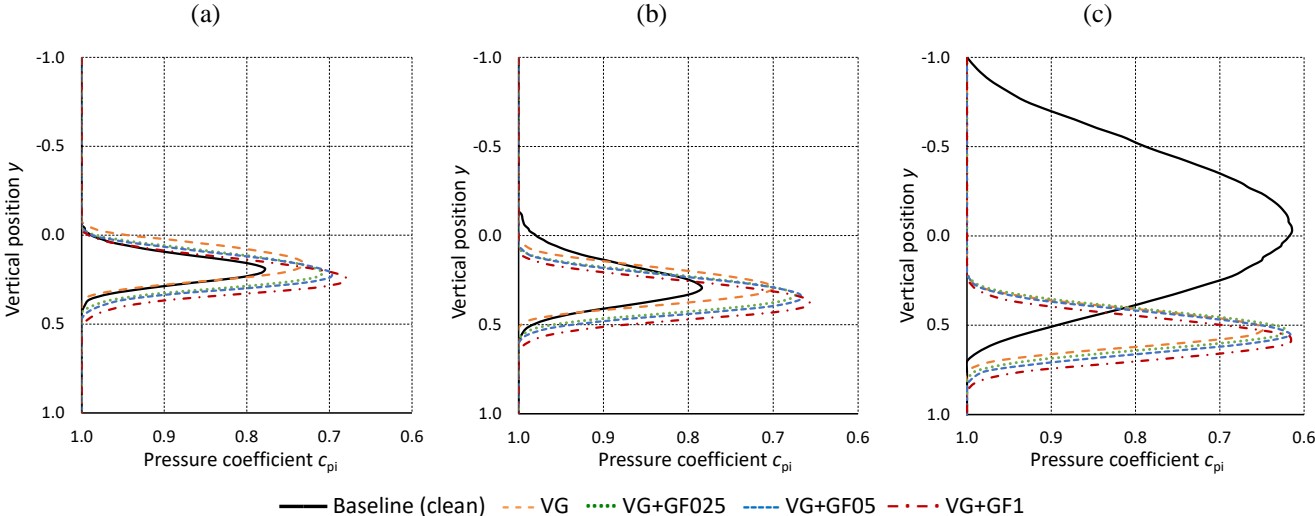

Figure 18. DU97W300. Pressure coefficients over vertical wake rake positions in the clean case. Vortex generators and Gurney flaps. (a) $\alpha_{opt} = 9.5°$. (b) $\alpha_{cl,max} = 12.6°$. (c) $\alpha = 16.5°$.

At $\alpha = 16.5°$, the baseline airfoil is clearly stalling, see Figure 18c. At this point, the wake consists of separated flow, i.e. 3D structures that cannot be determined accurately by means of wake rake measurements anymore. However, the VG + GF configurations suppress the formation of stall cells so that the flow remains attached, as described by Manolesos and Voutsinas

(2015). Finally, at $\alpha = 17.5°$, which is not displayed here, the flow separates abruptly leading to a steep decline of the lift curves, as illustrated in Figure 19a. These load excursions are perceptible in the form of strong mechanical vibrations of the set-up as well as a heavy roaring sound inside the wind tunnel. In summary, the wake deficit remains similar in shape and amount comparing the VG (only) to the combined configurations, again pointing towards a favorable wake interaction between VGs and MGFs.

Figure 19 displays the polar curves of all VG + GF configurations. For clarity, characteristic lift and L/D values are summarized in Table 8. The results of both the baseline and the VG (only) configurations are in very close agreement with reference measurements of the DU. Baldacchino et al. (2018) tested a comprehensive variety of VG configurations including $h_{VG} = 0.8\%c$ at $x_{VG} = 30\%c$ and $Re = 2 \cdot 10^6$. However, the tripped case consists of SuS (only) ZZ tape with $h_{ZZ} = 0.17$ mm at $x_{ZZ} = 5.0\%c$, which explains the elevated $L/D_{max,ZZ}$ compared to the current measurements, as detailed in Table 8. Next, The VG +

GF configurations show that stall is delayed by around 4° coupled with a significant $c_{l,max}$ increase, again, followed by an abrupt and steep drop at stall, see Figure 19a and c. Despite the improved drag behavior at elevated AoA, the combined drag penalty causes $L/D_{clean}$ to decrease significantly at low and moderate AoA. Overall, the results of the VG + MGFs are preferred over the VG + GF1 due to the beneficial L/D distribution, see Figure 19b. Under tripped conditions, all PFC devices are capable of maintaining L/D at low and moderate AoA. Moreover, $L/D_{max,ZZ}$ is increased as it is shifted by around 5° recovering

a large area of otherwise separated flow in the range of $7.4° < \alpha < 15.6°$, see Figure 19d. Again, VG + MGFs are the preferred configurations.





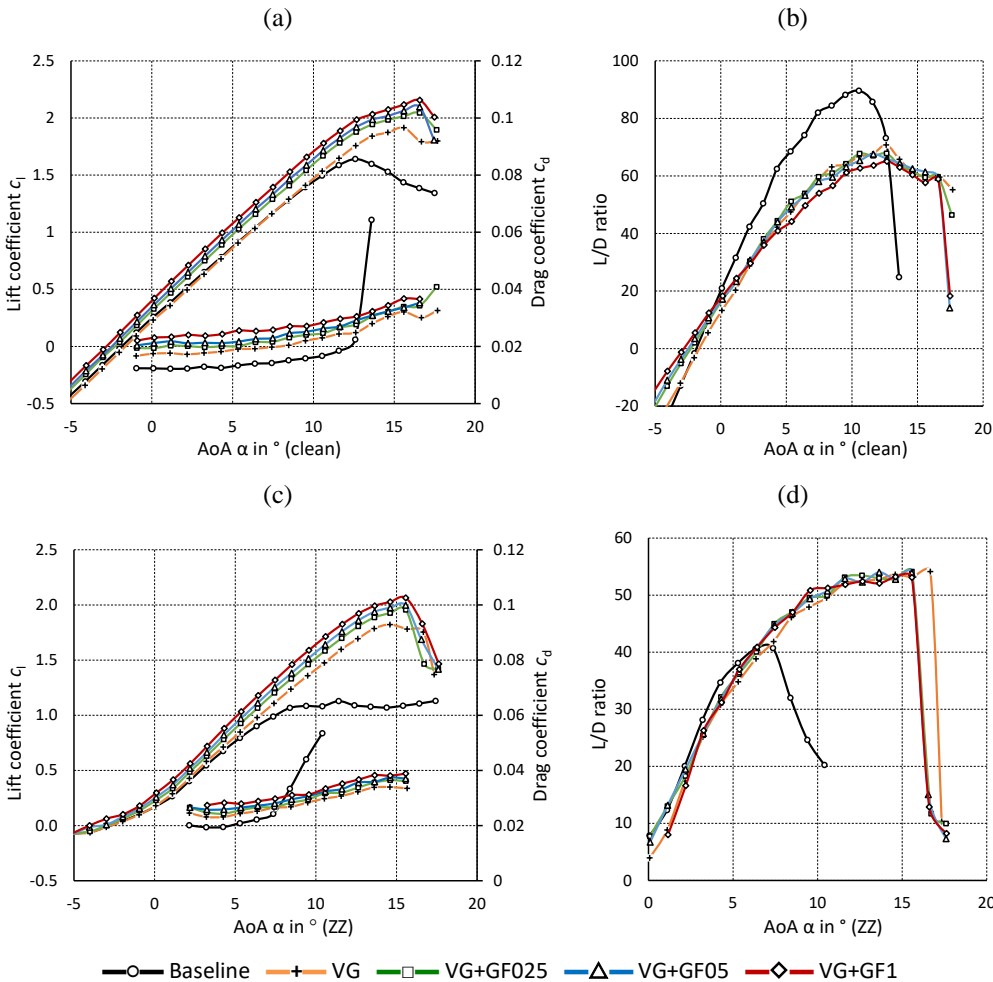

Figure 19. DU97W300. Vortex generators and Gurney flaps. (a) Lift and drag coefficients (clean). (b) L/D ratio (clean). (c) Lift and drag coefficients (ZZ). (d) L/D ratio (ZZ).

Table 8. DU97W300. Vortex generators plus Gurney flaps. Characteristic values. Reference data (ref.) is estimated based on Baldacchino et
al. (2018).

| | Clean | | Tripped | |
|---|---|---|---|---|
| | $c_{l,max}$ (α) | L/D$_{max}$ (α) | $c_{l,max}$ (α) | L/D$_{max}$ (α) |
| **Baseline** | 1.64 (12.6°) | 88 (9.5°) | 1.13 (11.4°) | 41 (7.4°) |
| **Baseline (ref.)** | 1.53 (12.4°) | 91 (9.3°) | 1.11 (9.2°) | 50 (6.2°) |
| **VG** | 1.91 (15.6°) | 71 (12.6°) | 1.82 (14.6°) | 52 (12.6°) |
| **VG (ref.)** | 1.95 (15.5°) | 73 (11.3°) | 1.87 (15.5°) | 60 (10.0°) |
| **VG+GF025** | 2.04 (16.6°) | 68 (12.6°) | 1.96 (15.6°) | 53 (12.6°) |
| **VG+GF05** | 2.10 (16.6°) | 66 (12.6°) | 2.00 (15.6°) | 52 (12.6°) |
| **VG+GF1** | 2.16 (16.6°) | 65 (12.6°) | 2.06 (15.6°) | 52 (12.6°) |





In summary, the DU97W300 measurements confirm the observations based on both the NACA63(3)618, see Sect. 4.1, and the AH93W174, see Appendix A: the VG is superposed by the GF effect, resulting in beneficial aerodynamic effects.

## 5. Rotor blade performance

The most beneficial combination of PFC devices is selected on the basis of the wind tunnel results, see Sect. 4. The experimental polar data is imported into the software QBlade (Marten, 2020) in order to create a generic rotor blade. The blade performance is simulated by means of two case studies, the retrofit application on an existing, and the new design application on a alternative rotor blade.

### 5.1 Blade configurations

As illustrated in Figure 5a, it is reiterated that the VG and the MGF configurations refer to $Re = 1.5 \cdot 10^6$, $h_{zz} = 0.3$ mm and 0.4 mm, respectively, $h_{VG} = 1.1$ %c and $h_{MGF} = 0.25$%c and 0.5%c, while $h_{GF} = 1.0$ %c is not considered relevant for this section of the report. Table 9 demonstrates the qualitative effect of the PFC devices on all tested airfoils by means of characteristic parameters.

Table 9. Performance evaluation of Mini Gurney flaps and vortex generators based on the wind tunnel tests of the NACA63(3)618, the AH93W174 and the DU97W300. ↑ for increase, ≈ for similar and ↓ for decrease.

|  | **Clean** | | | **Tripped** | | |
|---|---|---|---|---|---|---|
| **MGFs (only)** | $c_l\,(\alpha_{opt})$ ↑ | $\alpha_{cl,max}$ ≈ | $L/D_{max}$ ≈ | $c_l\,(\alpha_{opt})$ ↑ | $\alpha_{cl,max}$ ≈ | $L/D_{max}$ ↑ |
| **VG + MGF** | $c_l\,(\alpha_{opt})$ ↑ | $\alpha_{cl,max}$ ↑ | $L/D_{max}$ ↓ | $c_l\,(\alpha_{opt})$ ↑ | $\alpha_{cl,max}$ ↑ | $L/D_{max}$ ↑ |

The tendencies that are shown in Table 9 emphasize, first of all, that the effect of the PFC configurations is case dependent. In general, it is difficult to measure and to foresee the degree of LER, as described by Papi et al. (2021). For the purpose of this study, the principal objective is to improve the airfoil performance in the tripped case without jeopardizing the aerodynamic efficiency of the clean airfoil. Looking at the MGF (only) configurations, lift is increased at the design point and the stall behavior is basically consistent. The decambering effect of the ZZ tape is partly compensated for by the MGFs, as such improving the aerodynamic efficiency, while $L/D_{max,clean}$ is slightly improved or, at least, maintained. In the tripped case, VG + MGF achieve a triple improvement concerning lift increase, stall delay and aerodynamic efficiency. However, in the clean case, $L/D_{max,clean}$ decreases due to the combined drag penalty.

Apart from that, the principal objective of installing PFC devices on rotor blades is to enhance the energy yield over a life time of at least 20 years. Over time, the continuous development of adverse roughness effects is inevitable, as described by Maniaci et al. (2020) and Papi et al. (2021). Hence, in the mid to tip region, the driving parameters are the increase in both lift and



$L/D_{max}$. In the root to mid region, stall delay and $L/D_{max}$ are considered particularly important. Based on Table 9, the selected PFC configurations in both the clean and the tripped cases are summarized in Table 10.

Table 10. Selected PFC configurations for generic rotor blade design

| | |
|---|---|
| **NACA63(3)618** | GF025 |
| **AH93W174** | GF025 |
| **DU97W300** | VG + GF05 |


Concerning the NACA63(3)618 and the AH97W174, see Appendix A, GF025 is selected over GF05 as it is considered the more conservative option, especially in terms of $L/D_{clean}$ (α). In case of the DU97W300, VG + GF05 is the preferred option due to the significant stall delay as well as lift increase. In principal, both MGF configurations, i.e. GF05 and VG + GF025 and vice versa, lead to similar results, indicating a certain tolerance regarding the choice of the exact MGF height, as long as

Eq. ( 8 ) is fulfilled. It is noted that the AH93W174 is not included in the rotor blade design and performance simulations of the remaining Sect. 5.2 and 5.3.

## 5.2 Blade design

The experimental lift and drag polars are imported into the open source software QBlade. Figure 20 illustrates the rotor blade design of the NREL 5 MW reference wind turbine with $R = 63$ m, an average wind speed at hub height of $u_{avg} = 8$ m/s, and,

according to Eq. ( 23 ), a design tip speed ratio (TSR) of $\lambda_{opt} = 8$, as specified by Jonkman et al. (2009). The NREL blade is used as a template for the so-called generic rotor blade, which is designed in order to identify the impact of the PFC configurations on the rotor performance of an aerodynamically optimized blade. It is scaled down to $R = 20$ m, $u_{avg} = 7$ m/s and $\lambda_{opt} = 7$, as such reaching Re numbers closer to the wind tunnel tests, which are in the range of $1.5 \cdot 10^6$ to $2 \cdot 10^6$ rather than $3 \cdot 10^6$ to $9 \cdot 10^6$.

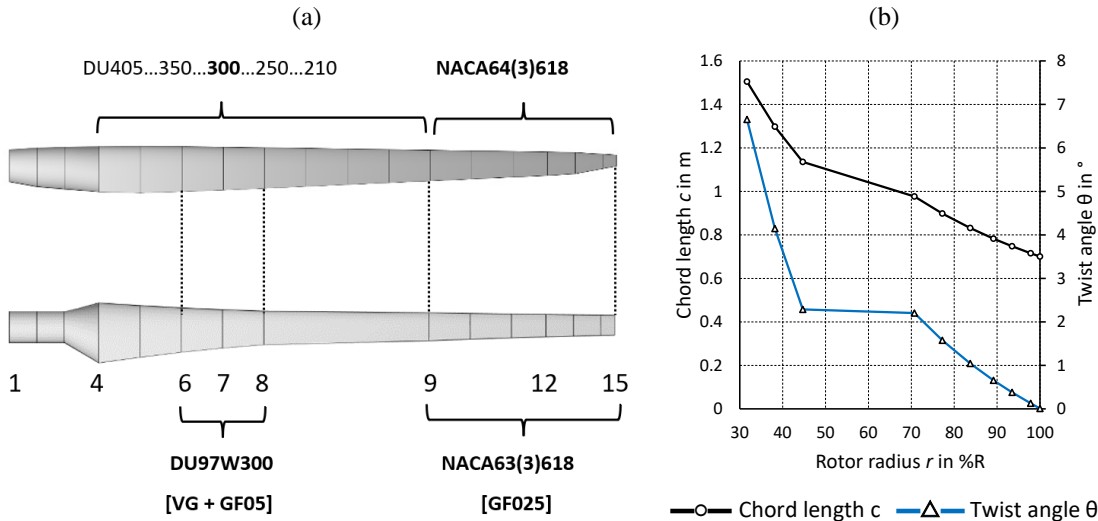



Figure 20. (a) Design of generic rotor blade based on the NREL 5 MW reference turbine. (b) Geometry of generic blade over rotor radius.

Figure 20a illustrates that the DU97W300 is placed from blade position 6 to 8 and the NACA63(3)618 between position 9 to 15. The mid span region is simply interpolated between position 8 and 9. Figure 20b shows the chord length $c$ $(r)$ and the twist angle $\theta$ $(r)$ of the generic blade, both of which are determined by means of the aerodynamic blade optimization procedure of Schmitz (1956), as described by Gasch and Twele (2012),

$$\lambda(r) = \frac{2\pi f r}{u_{avg}},$$

( 23 )

where $f$ is the rotational frequency in Hz and $\lambda(R) = \lambda_{opt}$.

$$c(r) = \frac{16\pi}{B} \cdot \frac{r}{c_l(\alpha_{opt}(r))} \cdot \sin^2\left(\frac{1}{3} \cdot tan^{-1}\left(\frac{R}{\lambda_{opt} \cdot r}\right)\right),$$

( 24 )

$$\theta(r) = \varphi(r) - \alpha_{opt}(r) = \frac{2}{3} \cdot \tan^{-1}\left(\frac{R}{\lambda_{opt} \cdot r}\right) - \alpha_{opt}(r),$$

( 25 )

where $B$ is the number of blades and $\varphi$ $(r)$ the inflow angle in °.

The Schmitz design, i.e. Eq. ( 24 ) and ( 25 ), leads to elevated chord lengths and twist angles in the root region due to the decreasing rotational speed and thus $\lambda$ $(r)$, towards the blade root. For practical and logistical reasons, the chord length in the root region, i.e. $r < 30\,\%R$, is usually reduced in order to restrict the volume and weight of large rotor blades. Hence, the numerical results of the generic blade are only considered to be feasible between position 6 at $r = 31.7\%R$ and the very tip.

Besides, no specific tip design is implemented.

Next, two generic case studies are defined and presented. The first one consists of the retrofit application of PFC devices that are installed on an existing rotor blade during regular maintenance activities. The original blade design is based on a smooth surface, i.e. clean airfoil polars. Over time, LER effects reduce the aerodynamic efficiency and thus the AEP. Subsequently,

VGs and GFs are installed in order to recover some of the decreased power output. The simulation parameters of the retrofit case, i.e. the blade geometry and all controller settings, are identical. The experimental polar data files of the clean, are replaced by the tripped baseline and, subsequently, by the tripped baseline including the PFC configurations, as illustrated in Figure 21.

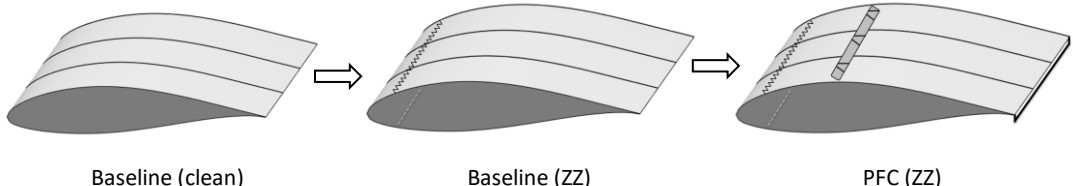

Figure 21. Retrofit application of passive flow control devices on a generic rotor blade section.

The second case study consists of the new design application of PFC devices that are installed as part of the blade

manufacturing process on the ground. The original, i.e. clean blade, is adversely affected by LER, as shown in Figure 21. The performance is compared to an alternative blade configuration including the PFC devices as part of the design process itself, see Figure 22.





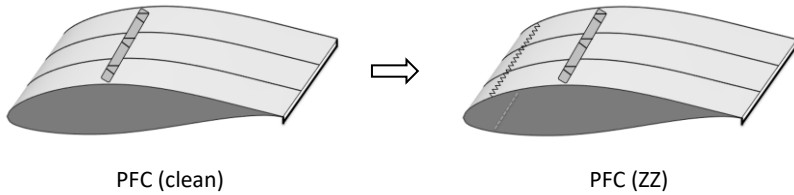

PFC (clean)                    PFC (ZZ)

Figure 22. New design application of passive flow control devices on a generic rotor blade section.

The blade geometry, $c\ (r)\ and\ \theta\ (r)$, is calculated separately for the clean baseline and the new blade including PFC devices.

Applying Eq. ( 24 ) and for $R$, $\lambda_{opt}$, $B$ to be identical, the ratio between the chord length at each blade position, $r$, is inversely proportional to the corresponding lift coefficients at the design AoA,

$$\frac{c\ (r)}{c_{PFC}(r)} = \frac{c_{l,PFC}\ (\alpha_{opt,PFC})}{c_l\ (\alpha_{opt})}. \qquad (\ 26\ )$$

According to Eq. ( 26 ), $c_{PFC}(r)$ decreases for larger $c_{l,PFC}$ values due to the MGF and VG effect, as illustrated in Figure 24.

### 5.3 Blade simulations

The rotor blade simulations are performed using the steady Blade Element Momentum (BEM) method, which is embedded
into QBlade, v99 based on Hansen (2015). The BEM simulation entirely depends on the quality of the imported polar data. Furthermore, empirical correction algorithms are activated, as described in Marten et al. (2013), including root and tip losses, thrust forces of heavily loaded rotors (Glauert correction) and spanwise crossflow effects. Apart from that, the results are interpolated between adjacent blade sections. The power curves are determined with respect to a rated power output of $P_{max} =$ 600 kW at $u_{rated} = 12$ m/s. In all cases, the basic pitch and rpm controller settings are optimized for reaching maximum power
output. Following Gasch and Twele (2012), the AEP or $E$ is calculated by means of the Weibull distribution using the probability factors $k = 2$ for a typical site in central Europe and the average wind speed at hub height, $u_{avg} = 7$ m/s over an operational wind speed range between $u_{cut-in} = 3$ m/s and $u_{cut-out} = 25$ m/s.

### 5.3.1 Retrofit application

The BEM results of the retrofit application are presented. Figure 23 shows the AoA along the local rotor radius, $r$. The clean
baseline coincides with $\alpha_{opt,clean}$ (L/D$_{max}$) = 9.5° of the DU97W300 and $\alpha_{opt,clean} = 6.4°$ of the NACA63(3)618. Replacing the clean by the tripped polar data, the AoA are significantly increased, see Figure 23a. In fact, the DU97W300 is stalling for $\alpha >$ 10.5°, so that the local L/D drops dramatically towards the root region, see Figure 23b. This adverse effect of the ZZ tape is partly compensated for by the PFC devices: the AoA are much closer to $\alpha_{opt,clean}$, so that $L/D\ (r)$ is partly recovered indicating that the PFC configuration is less sensitive to LER. Figure 23c shows the power coefficients over the complete operational
range of the rotor. In the tripped case, the $c_p$ curve is shifted towards higher TSR, so that $\lambda$ ($c_{p,max,ZZ}$) = 8 rather than $\lambda_{opt} = 7$. Hence, $c_{p,max,clean}$ ($\lambda_{opt}$) = 0.48 is decreased by 13 % to $c_{p,max,ZZ}$ ($\lambda_{opt}$) = 0.42. After the retrofit application, the $c_p$ curve is closer to the design point with $c_{p,max,PFC(ZZ)}$ ($\lambda_{opt}$) = 0.45, so that the relative decrease is reduced to 5.7 %.





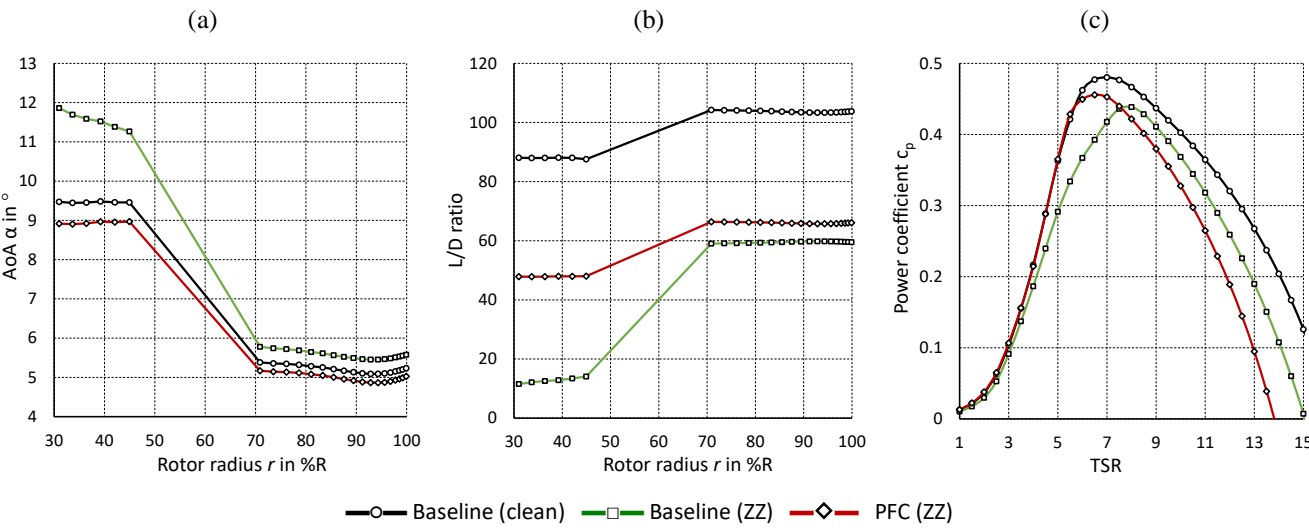

Figure 23. Rotor blade performance simulation of baseline and retrofit application. (a) AoA over rotor radius at $\lambda_{opt} = 7$. (b) L/D over rotor radius at $\lambda_{opt} = 7$. (c) Power coefficients over TSR.

Furthermore, the AEP amounts to $E_{clean} = 1594$ MWh, $E_{ZZ} = 1464$ MWh (-8.2 %) and $E_{PFC(ZZ)} = 1529$ MWh (-4.1 %). In this generic case study, the energy decline due to forced LE transition is approximately halved by the retrofit application of MGFs and VGs.

## 5.3.2 New design application

The BEM results of the new design application are presented. According to Eq. ( 26 ), the lift increase caused by the MGFs
leads to a significant chord length reduction, as illustrated in Figure 24a and b.

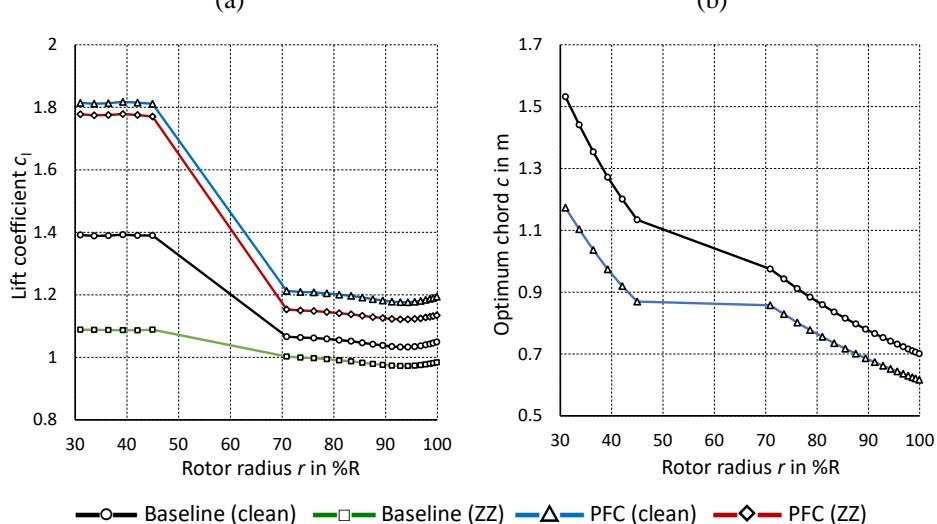

Figure 24. Blade geometry of baseline and new design application at $\lambda_{opt} = 7$. (a) Lift coefficients over rotor radius (b) Optimum chord length over rotor radius.



Compared to the baseline case, $c_{PFC}$ ($r$) is reduced by 23.4 % in the root and by 12 % in the tip region. Regardless of any structural-dynamic considerations, this approach potentially contributes to the development of more slender and thus lighter

blades. Apart from saving material costs, as previously suggested by Fuglsang et al. (2004), see Sect. 1.5, periodic gravitational load alternations and thus fatigue loads might be mitigated.

Next, Figure 25 shows the BEM simulation results. It is reiterated that the clean and tripped baseline configurations are identical to the retrofit application.

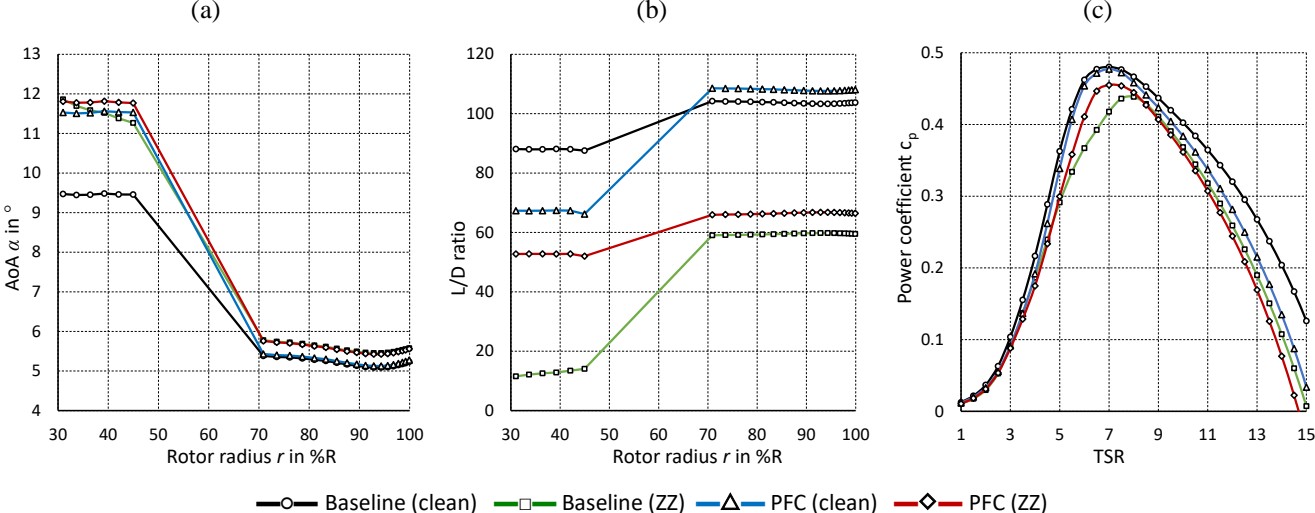

Figure 25. Rotor blade performance simulation of baseline and new design application. (a) AoA over rotor radius at $\lambda_{opt}$ = 7. (b) L/D over rotor radius at $\lambda_{opt}$ = 7. (c) Power coefficients over TSR.

Looking at the root region, the design AoA at $\lambda_{opt}$ = 7 is shifted from $\alpha_{opt,Base(clean)} \approx 9.5°$ to $\alpha_{opt,Base(ZZ)} \approx 11.5°$, where the DU97W300 is already stalling, see Figure 25a. The clean and the tripped PFC cases, on the other hand, lead to a significantly higher design AoA, i.e. $\alpha_{opt,PFC} \approx 12°$. Unlike the baseline configurations, stall is delayed by the VGs until $\alpha_{cl,max,PFC} \approx 16°$. In

the tip region, the MGF configuration leads to identical $\alpha$ ($\lambda_{opt}$) compared to the respective baseline. As a result, $L/D_{clean}$ ($r$) in the root region is decreased compared to the clean baseline due to the drag penalty of the PFC devices, see Figure 25b. However, in the tripped case, the reduction of $L/D_{PFC(ZZ)}$ ($r$) is significantly less severe compared to the baseline case because of stall delay. Moreover, looking at the tip region, the MGF effect leads to elevated $L/D$ ($r$) in both the clean and, particularly, the tripped case. Again, the $L/D$ curves indicate that the PFC configuration is less sensitive to LER. Figure 25c shows the $c_p$

curves. In the tripped baseline case, the design TSR is shifted to $\lambda(c_{p,max,ZZ})$ = 8 rather than $\lambda_{opt}$ = 7. For both PFC cases, $c_{p,max,PFC}$ remains at $\lambda_{opt}$ = 7, so that the overall shape of the clean baseline curve is maintained. As a consequence, $c_{p,max,PFC(clean)}$ = 0.48 is almost identical to the clean baseline, despite moderate differences at elevated TSR, i.e. for $\lambda > \lambda_{opt}$. Furthermore, $c_{p,max,PFC(ZZ)}$ ($\lambda_{opt}$) = 0.45 decreases by only 4.6 % rather than 13 % compared to the respective clean case. The AEP amounts to $E_{Base(clean)}$ = 1594 MWh and $E_{Base(ZZ)}$ = 1464 MWh (-8.2%), compared to $E_{PFC(clean)}$ = 1587 MWh and $E_{PFC(ZZ)}$ = 1536 MWh (-3.6%).



Hence, the difference between the clean baseline and clean PFC configuration is minor. In addition, the blades are more slender and the energy decline due to forced LE transition is mitigated by the new design application of MGFs and VGs.

## 6 Conclusions

The present study investigates the use of MGFs and their combination with VGs for improved rotor blade performance of HAWTs. The main conclusions are summarized alongside each sections of this report.

1. The aerodynamic effects of either GFs or VGs on the airfoil performance is well documented in literature. Various wind tunnel studies emphasize the benefits of very small GFs, so-called MGFs, particularly in terms of the aerodynamic efficiency. However, the simultaneous installation of GFs and VGs is less profoundly researched.

2. The simulation software XFOIL is used to determine the appropriate dimensions of each PFC device in relation to the boundary layer displacement thickness of the following airfoils: the NACA63(3)618, the AH93W174 and the
DU97W300. The definition of a MGF refers to an effective height in the range of one to two times the local boundary layer displacement thickness at the design AoA. Within this boundary condition, the MGF effect on the airfoil performance appears to be beneficial in terms of the lift performance, the stall behavior and the aerodynamic efficiency.

3. The baseline results of the tested airfoils are validated against literature data. The effects of the PFC devices are
captured successfully by the current measurement methods, i.e. the force balance for the lift, and the wake rake for the drag coefficients. Furthermore, the wake rake registers both the increased downwash angle due to the GFs and the stall delay due to the VGs. Compared to the clean baseline, the additional intensity of the wake unsteadiness diminishes in relation to the GF height, i.e. using MGFs.

4. The impact of MGFs and VGs depends on whether the free or the forced boundary layer transition is applied. MGFs
(only) increase the pre-stall lift performance compared to the corresponding baseline. The aerodynamic efficiency is maintained in the clean, and improved in the tripped case. Combining MGFs and VGs leads to stall delay, lift increase and a significant improvement of the aerodynamic efficiency in the tripped case. However, in the clean case, maximum L/D is decreased due to the combined drag penalty. In summary, the selected configurations include the smallest MGF height of 0.25 %c on both the NACA63(3)618 and the AH93W174, and the medium sized MGF height
of 0.5 %c plus VGs with a height of 1.1 %c on the DU97W300.

5. The experimental polar data is imported into the software QBlade in order to create an aerodynamically optimized rotor blade. The performance of the generic rotor blade is simulated based on two case studies, the retrofit application on an existing, and the new design application on a alternative rotor blade. The retrofit application alleviates the adverse effects of the ZZ tape. Separation is delayed in the root to mid region, and the aerodynamic efficiency and
thus power output, is recovered in the mid to tip region of the blade. The new design application leads to a more




slender blade while maintaining the power output. Furthermore, the alternative blade appears to be more resistant against the effects of forced leading edge transition.

Further research on MGFs and their interaction with relatively small VGs is recommended, especially considering leading
edge roughness effects. Next steps involve the design of Mini VGs in conjunction with Mini GFs to further reduce drag. Moreover, advanced methods for emulating different degrees of leading edge roughness should be applied. In addition, a complete aeroelastic simulation of the rotor performance is required, especially with regard to open field tests of MGFs in combination with VGs on large wind turbine rotor blades.

## Appendix A: AH93W174

The wind tunnel results of the AH93W174 are presented alongside the NACA63(3)618 and the DU97W300, see Sect. 4. For brevity, only the L/D graphs and the characteristic lift and L/D values are included in A2 and A3.

### A1. Baseline

The clean and the tripped polar curves are presented. Stall is initiated at $\alpha_{cl,max} = 10.5°$ and $\alpha_{opt,clean} = 8.5°$ decreases to $\alpha_{opt,ZZ} = 5.3°$, see Figure A1a. Figure A1b shows that the aerodynamic efficiency drops from $L/D_{max, clean} = 118$ to $L/D_{max,ZZ} = 61$. For
clarity, characteristic lift and L/D values are summarized in Table A 1.

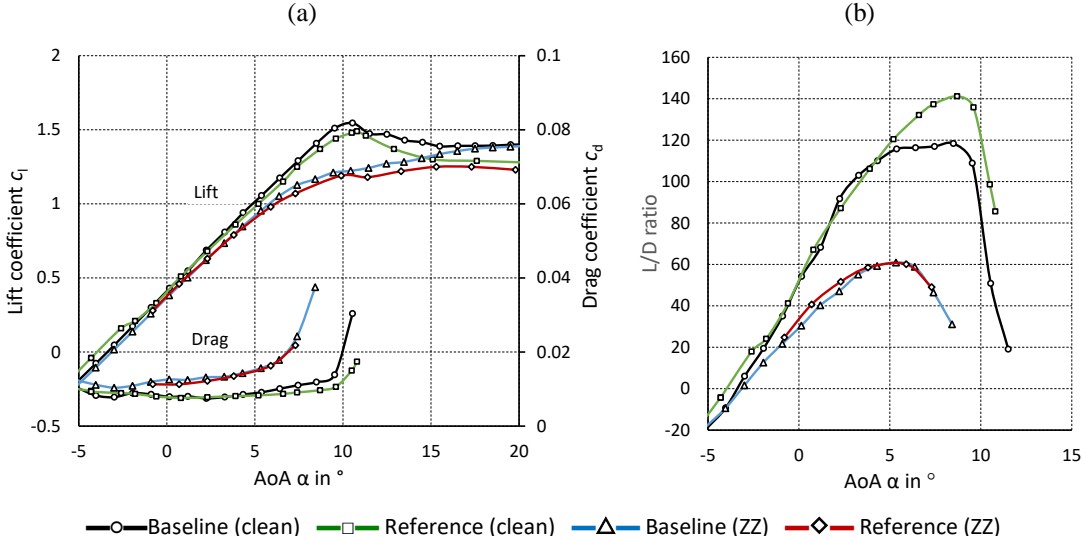

Figure A1. AH93W174. Clean and tripped baseline cases at $Re = 1.5 \cdot 10^6$ compared to reference data from Althaus (1996) at $Re = 1.5 \cdot 10^6$. (a) Lift and drag coefficients. (b) L/D ratio.

The baseline measurements are compared to literature data from the airfoil catalogue of Althaus (1996). The AH93W174 was tested in the laminar wind tunnel of the IAG at Stuttgart University. The inflow turbulence in the open return test section was



0.012 %. Lift and drag were determined at $Re = 1.5 \cdot 10^6$ using the methods that were previously mentioned in case of the NACA63(3)618, see Sect. 4.1.1. The wake rake was positioned 30%c downstream the airfoil TE. Forced LE transition was triggered at the SuS (only) via trip wire using a diameter of 0.3mm at $x_{SuS} = 5.0$ %c.

Figure A1a shows very good agreement between both lift and drag curves, despite slight deviations approaching $c_{l,max}$.
Furthermore, the clean case shows a steeper drag increase at elevated AoA resulting in reduced $L/D_{max,clean}$ compared to the reference data, see Figure A1b. The reason is probably the inflow turbulence, which is significantly stronger in case of the current measurements, hence leading to earlier transition at elevated AoA and thus higher drag values. Moreover, the difference in the rake positions downstream the airfoil TE, i.e. 30%c versus 100%c, might contribute to the deviations in the drag results.

**A2. Mini Gurney flaps**

The polar curves of all GF configurations are presented. Overall, the results agree with the NACA63(3)618 measurements, see Sect. 4.1.2. For brevity, only the $L/D$ graphs and the characteristic lift and $L/D$ values are shown. According to Figure A 2 and Table A 1, the preferable results are achieved for GF025 in the clean, and GF05 as well as GF1 in the tripped case. Again, the beneficial GF effect is more pronounced in the tripped cases.

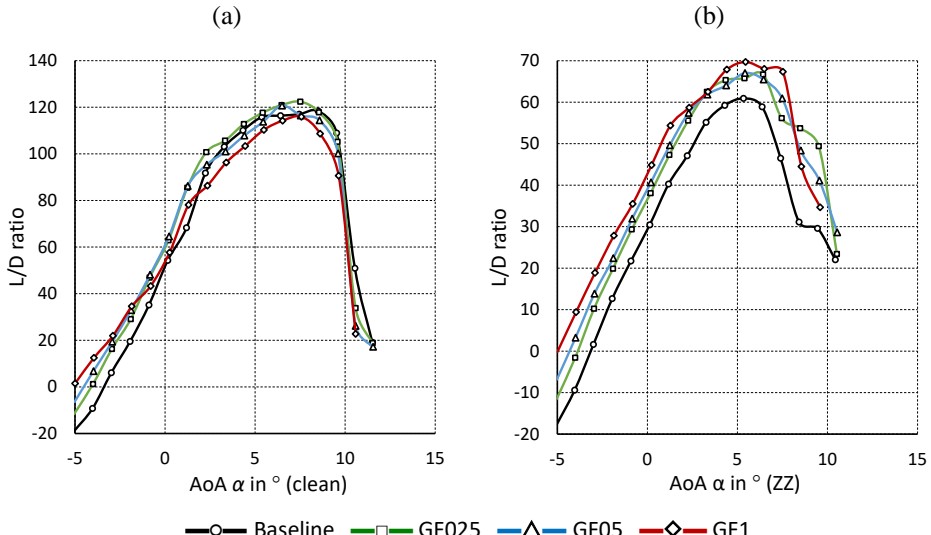

Figure A 2. AH93W174. Gurney flaps. (a) $L/D$ ratio (clean). (b) $L/D$ ratio (ZZ).

Table A 1. AH93W174. Gurney flaps. Characteristic values.

| | AH93W174 | | | |
|---|---|---|---|---|
| | **Clean** | | **Tripped** | |
| | $c_{l,max}$ (9.5°) | $L/D_{max}$ (7.5°) | $c_{l,max}$ (10.5°) | $L/D_{max}$ (5.4°) |
| **Baseline** | 1.51 | 117 | 1.22 | 61 |





| | | | | |
|---|---|---|---|---|
| **GF025** | 1.66 | 122 | 1.37 | 66 |
| **GF05** | 1.73 | 116 | 1.44 | 67 |
| **GF1** | 1.83 | 116 | 1.55 | 70 |

## A3. Vortex generators plus Mini Gurney flaps

The polar curves of all VG + GF configurations are presented. Overall, the AH93W174 results agree with the NACA63(3)618 measurements. In fact, the beneficial effects of both the VG (only) and the VG + GF configurations are significantly more pronounced: $\alpha_{cl,max}$ is shifted by around 6° and $\alpha_{opt,ZZ}$ by approximately 5°, so that a large area of otherwise separated flow is recovered at elevated AoA. Hence, in both cases, the preferred results are achieved using VG + GF025. For brevity, only the L/D graphs and the characteristic lift and L/D values are presented in Figure A 3 and Table A 2, respectively.

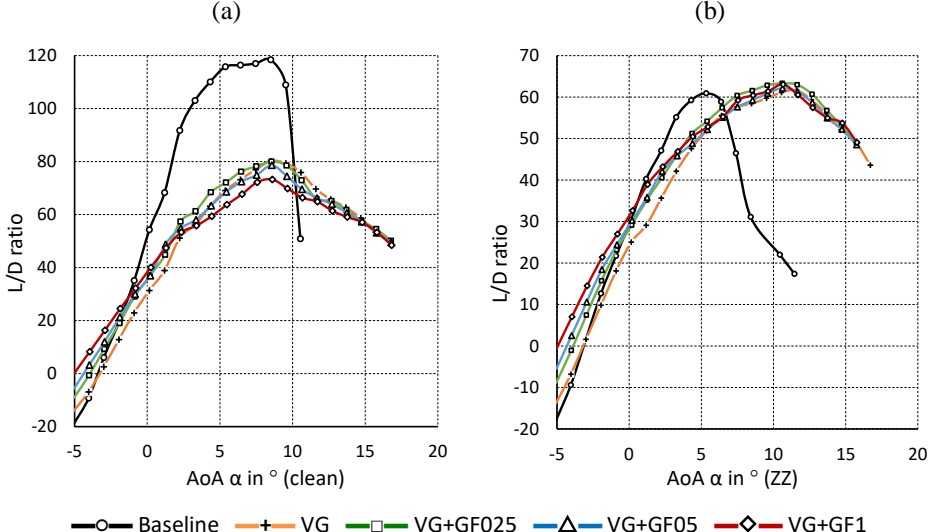

Figure A 3. AH93W174. Vortex generators and Gurney flaps. (a) *L/D* ratio (clean). (b) *L/D* ratio (ZZ).

Table A 2. AH93W174. Vortex generators combined with Gurney flaps. Characteristic values.

**AH93W174**

| | **Clean** | | **Tripped** | |
|---|---|---|---|---|
| | $c_{l,max}(\alpha)$ | $L/D_{max}(\alpha)$ | $c_{l,max}(\alpha)$ | $L/D_{max}(\alpha)$ |
| **Baseline** | 1.55 (10.5°) | 118 (8.5°) | 1.55 (10.5°) | 61 (5.3°) |
| **VG** | 2.05 (16.7°) | 80 (8.5°) | 2.05 (16.7°) | 61 (10.6°) |
| **VG+GF025** | 2.18 (16.7°) | 80 (8.5°) | 2.11 (15.8°) | 63 (10.6°) |
| **VG+GF05** | 2.25 (16.7°) | 79 (8.5°) | 2.19 (15.8°) | 62 (10.6°) |
| **VG+GF1** | 2.35 (16.7°) | 73 (8.5°) | 2.28 (15.8°) | 63 (10.6°) |

**Data availability.**

Measurement data and results can be provided by contacting the corresponding author.

**Author contribution**

JF designed and fabricated the wake rake and the vortex generators. JA validated the wake rake measurements and designed the Mini Gurney flaps. JA, MM, GWD, JF and AS prepared and conducted the wind tunnel experiments. JA performed the airfoil and rotor blade simulations. JA and AS processed the experimental and numerical data. JA wrote the manuscript with
the support of all co-authors, who contributed with important suggestions.

**Competing interests**

The authors declare that they have no conflict of interest.

**Acknowledgements**

The authors would also like to appreciate the constant support of the technicians of the Hermann-Föttinger Institut at the
Technische Universität Berlin.

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
