# Peer review of "Experimental investigation of mini Gurney flaps in combination with vortex generators for improved wind turbine blade performance"

_Wind Energy Science, 2021_

## Referee Comment (RC1)

**Journal:** WES
**MS No.:** wes-2021-124
**Title:** "Experimental investigation of Mini Gurney Flaps in combination with vortex generators for improved wind turbine blade performance "

**Author(s): Jörg Alber, Marinos Manolesos, Guido Weinzierl-Dlugosch, Johannes Fischer, Alexander Schönmeier, Christian Navid Nayeri, Christian Oliver Paschereit, Joachim Twele, Jens Fortmann, Pier Francesco Melani[5], and Alessandro Bianchini**

**General comments:**

This paper present wind tunnel investigations on the effects of Mini Gurney flaps (MGFs) and their combination with vortex generators (VGs) on the performance of airfoils and wind turbine rotor blades.

This paper present high quality experiments with a lot of details on how to design the combined configurations of MGF and VG. Output results on the efficiency of passive devices should however be taken with caution because of two main reasons:
- the zztape effects are more important than passive device effects which is certainly an important issue for real blades that have generally LE erosion during operation. Same studies with different LE roughness should certainly be performed as pointed out by the authors.
- experiments were performed in a low turbulent intensity wind tunnel facility, which is far from the environment of operated blades and may lead to a decrease of actuator efficiency.

An important output of the present paper is the new design opportunities (chord length significantly reduced) that is provided using these passive devices. This study explains to do such a new blade design  (the first detailed paper on that matter from my knowledge).

It has however some issues that need to be corrected. The major issues concern :
- the scientific objective that is unclear regarding the available literature of section 1.5.
- the hypothesis on the design of MGF and Vgs  that are not always formulated, especially when using Xfoil to design actuators at high angle of incidence.

I thus recommend the publication of the paper with corrections detailed below.

**MAJOR ISSUES:**

**Q1:** P6L137 How the present study is original from the existing literature of section 1.5 ?

**Q2:** p8L190: "relative strong turbulence intensity of Ti=0.3%"
For atmospheric flows in which wind turbine operates, the turbulence intensity is rather around 10%, please remove strong and put it in the context of wind tunnel facilities.

**Q3:** P9L216: this "The model revealed a non-proportional dependency on the GF height"
contradict this "diminishing HGF, dCL/hGF increased, whereas dCd/hGF decreased.
Also, from equation 6: Cl/Cd ~ dCl/Cd , which also contradicts the non-proportional dependency …

It is not clear what is non-proportional to what ?

**Q4:** P9L221: hGF is  defined relatively to the blade chord while the conclusion is "GF needs to be submerged deeply into the local BL"
The height of the device should be expressed relatively to the boundary layer thickness to do that conclusion. Also, it can't be reduced to the boundary layer thickness dependency only, as the boundary layer is never in equilibrium on blades but subjected  to different pressure gradient history depending on the blade shape. The conclusion of Alber et al (2017) study is therefore limited to the tested configurations.

**Q5:** P9L224: "ratio between the GF height and the BL displacement thickness at the TE"
Why at the TE ?
This is rather at the location of the GF. The effect of GF location is certainly another parameter that needs to be explored.

**Q6:** P9L225: It is not clear here why MGF is designed at the optimal angle of incidence ? It is certainly the angle of incidence corresponding to the maximum TE boundary layer thickness, so the MGF design is detrimental to other angles of incidence ?
Please explain.

**Q7:** P10L2: "hMGF < delta*", the impact of MGF on the airfoil performance becomes insignificant.
Even if it seems obvious that the MGF size has some low limitation, how do you end-up with this value ?

**Q8:** p10L228: "hMGF ~ 0.25 delta"
Why chosing ¼ while 2/3 would lead to an higher aerodynamic impact and is stil compatible to eq. 8 ?

**Q9:** p10L234: "0.1%c < hMGF < 0.7%c"
According to table 3, the minimum value of delta is 0.82%c, leading to hMGF=0.2%c (according to equation 9). Please correct.

**Q10:** P10L239: "All tested … in relation to the size of GF", this sentence is not clear, is the GF height varying from 0.3mm to 0.6mm ? It does not seems so for the smallest MGF as the chord is c=0.6m and hMGF=0.25%c=0.15mm. Please make it clearer.

**Q11:** P10L241: on the vortex generators design
The BL transition can be extracted from Xfoil. However, Xfoil is known to be limited to attached flow configurations with difficulties to correctly predict forces when there exist flow separation and especially at Clmax. I also don't know any Xfoil output on the mean separation line.

It is therefore not clear here how the location of the mean separation line is obtained ?

 Please be clearer.

**Q12:** Also, once the flow is separated, there is not anymore a boundary layer flow, so the standard boundary layer thickness definition fails. Please specify how do you define it ?

**Q13:** P11L1: "at stall, delta is similar in both the clean and tripped cases"
Stall configuration refers to a full flow separation over the blade suction side, so no delta can be measured (as there no boundary layer anymore).

Please reformulate to be clearer.

**Q14:** P17L377: From

Dan H. Neuhart and Catherine B. McGinley "Free-Stream Turbulence Intensity in the Langley 14- by 22-Foot Subsonic Tunnel" NASA report NASA/TP-2004-213247, Langley Research Center, Hampton, Virginia

The turbulent intensity is one order of magnitude higher, between ~0.07% to 0.08%.

There is certainly a mistake in the reported Ti (0.005%), please correct.

**Q15:** The drag signal is acquired at 10kHz and the lift is acquired at 5kHz, it is therefore possible to plot the standard deviation with the AOA. Please add this quantity that will help to evaluate further the actuator efficiency.

**Conclusion:**
The ZZ tape has more impact on the L/D ratio than the actuators themselves. Therefore authors raise naturally the question of the blade roughness impact on their conclusion, but should also raise the question of the turbulent intensity impact, that has the ability to enhance the mixing rate of separated shear layers near the maximum lift values (or near stall).

**MINOR ISSUES:**

**Q1:** This citation is not a peer review journal nor a conference paper:

Schatz, M., Günther, B., and Thiele, F.: Numerical Simulation of the Unsteady Wake behind Gurney-Flaps, available at: https://www.cfd.tu-berlin.de/research/flowcontrol/gurneys_en/ (last access: 17 October 2020), 2004a.

**Q2:** L93: "the unsteadiness vanished as the AoA is increased from 0 to 4°"
why only until 4° ? It increases again afterwards ?

**Q3:** P2L37: what PFC stands for ?

**Q4:** P12L267: what is the purpose of the "airfoil box" ?

**Q5:** P12L274: from the static pressure difference between the inlet and outlet of a duct you only get the pressure losses, not the dynamic pressure used to measurement the inflow velocity. Please correct the sentence.

**Q6:** P12L276: How the blade is attached to the balance system ?

**Q7:** P12L277: What are the wind tunnel boundary layer thickness value ? And what are the end plates dimensions and how they were chosen ?

**Q8:** P12L277-278: The flow in the bypass region is certainly very complex with interaction between the facility boundary-layer and the end plate boundary layer. Why using that location as a reference velocity ? Is the prandtl tube in front of the blade sufficiently insufficient ? Is there blade induction effects at this location ?

**Q9:** P14L1: what is the origin of x ? because x=c from figure 4a is at the trailing edge … I guess you mean x/c=2 ?
Please correct.
Idem p25L494: 100%c → x/c=2

**Q10:** P14L2: what do you mean by "return to static pressure level in the wind tunnel" ?
If the pressure is measured in the blade wake area at high angle of incidence, 1D is certainly not sufficient to return to the static pressure level in the wind tunnel. Please be more specific.

**Q11:** P14L311: "two Prandtl tubes that are installed inside the downstream plane of the rake, one on top and one below the casing"
This sentence is not clear as Prandtl tubes measure the dynamic pressure (difference between static and total pressure). Please be clearer.

**Q12:** P14: careful on the notations: deltapP(yi)=deltaPi  of equation 11 ?
Ptotal = Pbar total (the question behind is: how long the signal was acquired and does the bar mean averaged over that signal ?)
Pstatic=Pbar static (idem)?

**Q13:** P14L330: please explain (quickly) where the relation 14 comes from.

**Q14:** P14L340: About Lambda: is it related to the airfoil thickness (and so dependent on each airfoil), or is it the height of the test section (and so a fixe value). Please make it clearer.

**Q15:** Please provide the raw lift and drag curves to see the improvement when applying the wind tunnel correction.

**Q16:** P18L396: Next, figure 12a … isn't it figure 11a ?
 Please check.

**Q17:** P18 legend of figure 11 on p19.
Please correct

**Q18:** p21L1-2: it should also be noted that there is only marginal L/D improvement between GF cases.

**Q19:** P26L506: normalizing the vertical position with the chord and locate the center relatively to the blade will help to evaluate the wake extend and thus conclude on the flow separation extend, that is not necessarily going up to the trailing edge - stalled (in figure 15a: the baseline zz case indicates that the flow separation do not occurs at the leading edge).

**Q20:** P26L509: please replace "suppress" by "delay"

**Q21:** p27 Figure 19: I don't see the benefit of adding GF (or MGF) compared to VG alone, and it is not commented in the article.
Please explain why MGF+VG is better than VG alone for that configuration ?

**Q22:** P30: legend of figure 20 is misplaced.

---

## Referee Comment (RC2)

[revised manuscript text omitted]

<table>
<tr><td colspan="5" align="center">AH93W174</td></tr>
<tr><td></td><td colspan="2" align="center">Clean</td><td colspan="2" align="center">Tripped</td></tr>
<tr><td></td><td>$c_{l,max}$ ($\alpha$)</td><td>$L/D_{max}$ ($\alpha$)</td><td>$c_{l,max}$ ($\alpha$)</td><td>$L/D_{max}$ ($\alpha$)</td></tr>
<tr><td>Baseline</td><td>1.55 (10.5°)</td><td>118 (8.5°)</td><td>1.55 (10.5°)</td><td>61 (5.3°)</td></tr>
<tr><td>VG</td><td>2.05 (16.7°)</td><td>80 (8.5°)</td><td>2.05 (16.7°)</td><td>61 (10.6°)</td></tr>
<tr><td>VG+GF025</td><td>2.18 (16.7°)</td><td>80 (8.5°)</td><td>2.11 (15.8°)</td><td>63 (10.6°)</td></tr>
<tr><td>VG+GF05</td><td>2.25 (16.7°)</td><td>79 (8.5°)</td><td>2.19 (15.8°)</td><td>62 (10.6°)</td></tr>
<tr><td>VG+GF1</td><td>2.35 (16.7°)</td><td>73 (8.5°)</td><td>2.28 (15.8°)</td><td>63 (10.6°)</td></tr>
</table>

[revised manuscript text omitted]

---

## Author Comment (AC2)

[revised manuscript text omitted]

---

## Author Response (AR1)

[revised manuscript text omitted]

**Commented [D25]:** RC1(Q6minor): The attachment of the wing model to the force balance was specified

The inlet and the outlet of the duct walls are equipped with a ring line of pressure taps. The inflow velocity is determined from the contraction ratio and the static pressure difference. The airfoil model, or wing, is positioned in the centre of the test section, as displayed in Figure 6a. It is enclosed by two 1.5m long end plates that are parallel to the tunnel, as such reducing the influence of the wall BL. The velocities inside the 0.25 m wide bypass channels are measured via two separate Prandtl tubes to obtain the effective inflow velocity. The wing is directly clamped to the platform of the permanently installed force balance underneath the test section, see Figure 6b. Hence, the suspension is decoupled from the end plates and the tunnel walls. The AoA is controlled by means of a stepping motor with an accuracy of 0.1°, which is directly attached to the suspension. The wing models were CNC milled from a solid block of Obomodulan™, as described by Pechlivanoglou (2013). The chord length is 0.6 m and the span is 1.54 m resulting in an aspect ratio of 2.56.

**Commented [D26]:** RC1 (Q5minor): the measurement principle of the inflow velocity was specified.

RC1 (Q7minor): the dimensions of the end plates was included

RC1 (Q8minor): the reference Prandtl tube in front of the airfoil model was removed to avoid confusion

**Commented [D27]:** RC1(Q6minor): idem

**3.2 Measurement methods**

**3.2.1 Force balance**

[revised manuscript text omitted]

> **Commented [D31]:** RC1 (Q12minor): the data reduction process was explained with more clarity

$$c_{d,raw}(\alpha) = 2 \int \left( \sqrt{\frac{\Delta p_i}{\Delta p_{ref}}} - \frac{\Delta p_i}{\Delta p_{ref}} \right) \frac{dy}{c}, \tag{7}$$

> **Commented [D32]:** RC1 (Q13minor): The basic momentum loss equation that the scrip is based on was included

where, $\Delta p_i$ is the mean differential pressure value in Pa at each rake tube,

$$\Delta p_i = \Delta p(y_i) = \Delta p_{total}(y_i) - \Delta p_{static}, \tag{8}$$

and $\Delta p_{ref}$ the reference pressure in Pa of the free flow that is taken from the two uppermost and the two lowest rake tubes,

$$\Delta p_{ref} = 0.25 \cdot (\Delta p_1 + \Delta p_2 + \Delta p_{57} + \Delta p_{58}). \tag{9}$$

The pressure coefficient, $c_{pi}$, is defined as,

$$c_{pi} = \frac{\Delta p_i}{\Delta p_{ref}}, \tag{10}$$

Based on Eq. ( 7 ), the uncorrected drag contribution of each rake tube, $c_{di}$, becomes,

> **Commented [D33]:** RC1(Q13minor): idem

$$c_{di} = \sqrt{c_{pi}} - c_{pi}. \tag{11}$$

The uncorrected total drag coefficient is then numerically integrated over the spacing between the rake tubes using the trapezoid rule,

$$c_{d,raw}(\alpha) = \frac{1}{c} \sum_1^{58} (c_{di} + c_{di+1}) \cdot (y_{i+1} - y_i), \tag{12}$$

where $c$ is the airfoil chord length and $y_i$ the normalized position of each rake tube, as illustrated in Figure 7b.

The measured lift and drag polars, $c_{l,raw}(\alpha)$ and $c_{d,raw}(\alpha)$, are affected by the wind tunnel walls. The reasons are, first of all, that the solid blockage effect leads to the constriction of the curved streamlines around the airfoil model. Secondly, the wake blockage effect causes the constriction of the curved streamlines in the wake. For the results to be comparable to equivalent open flow conditions, it is necessary to apply wind tunnel corrections, as detailed in Appendix A. In the remainder of this report, the polar data refers to the corrected lift and drag coefficients, $c_l(\alpha)$ and $c_d(\alpha)$.

> **Commented [D34]:** RC2: Sect. 3.2.3. Wind tunnel corrections was shifted to Appendix A to shorten the main part of the report

> **Commented [D34]:** RC2: Sect. 3.2.3. Wind tunnel corrections was shifted to Appendix A to shorten the main part of the report

**3.3 Test matrix**

The inflow velocity, $u = 40$ m/s, corresponds to a Reynolds number of approximately $1.5 \cdot 10^6$. The free stream turbulence intensity of the empty wind tunnel is estimated by means of a Prandtl tube and is less than 0.3 %. The AoA ranges from $-5° < \alpha < 17°$ in steps of 1°. At each static AoA, there is a buffer of 4 s for the flow to settle, after which data is recorded for another 5 s. Hence, the total number of samples is $n = 5 \cdot 10^4$ for each rake sensor and $n = 2.5 \cdot 10^4$ for the load cells of the force balance. Before each test run, all sensors are subjected to a zero-offset measurement at standstill in order to reduce experimental errors. The sequence of measurements starts with the clean baseline followed by the three GF configurations, GF025, GF05 and GF1,

310 which refer to a GF height of 0.25 %c, 0.5%c and 1%c, respectively. Next, GF1 is removed and the VG array is installed, followed by the combined configurations, VG + GF025, VG + GF05 and VG + GF1. In the next round, ZZ tape is attached and the test matrix is repeated. Each complete cycle, clean and tripped, is measured within less than 24 hours for the environmental conditions to remain as constant as possible.

**4. Experimental results**

315 The presentation of the wind tunnel measurements is focused on the NACA63(3)618 with GFs and the DU97W300 including VGs plus GFs. All results refer to the clean and the tripped cases. They are presented in the form of both the polar curves and the wake pressure fields.

**4.1 NACA63(3)618: Gurney flaps**

**4.1.1 Polar curves**

325 Figure 8 shows the clean and the tripped polar curves of the NACA63(3)618. For clarity, characteristic lift and $L/D$ values are summarized in Table 6**Error! Reference source not found.**.

In the baseline cases, the drag results are valid until stall at $\alpha_{cl,max} = 10.5°$ and the lift curves are measured until the post-stall AoA of 18.5°, see Figure 8a and c. As expected, ZZ tape with $h_{ZZ} = 0.4$ mm manifests itself in a lift decrease, coupled with a significant drag increase. The design point declines from $\alpha_{opt,clean} = 6.4°$ to $\alpha_{opt,ZZ} = 5.4°$ and the corresponding aerodynamic

330 efficiency drops from $L/D_{max,clean} = 109$ to $L/D_{max,ZZ} = 60$, see Figure 8b and d. The clean and the tripped GF configurations are characterized by an increase in both lift and drag throughout the complete pre-stall region. Furthermore, the shape of the polar curves and the stall behaviour is maintained. In the clean cases, $L/D_{max}$ is only marginally improved by GF025 and GF05. Nonetheless, the significant lift increase is expected to be beneficial in terms of the rotor blade performance, as long as $L/D$ ($\alpha$) is maintained. As such, GF025 provides the preferred results, while GF1 leads to an overall $L/D$ ($\alpha$) decrease. In the tripped

335 cases, the aerodynamic efficiency is improved independently of the GF height. The reason is the significant expansion of the BL due to forced LE transition, so that larger GFs appear to be more beneficial.

**Commented [D36]:** RC2: the designation of the configurations (GF025, etc) were specified with more clarity.

**Commented [D37]:** RC2: The structure of this chapter was changed to shorten the report. For that, the results were restricted to

1.) the NACA63(3)618 with GFs (only) and

2.) the DU97W300 with VGs plus GFs

The following parts were moved to the appendices (pls note that the numbering of the sections and Figures has changed as they refer to the previous version of this manuscript):
- Fig. 10 + text
- 4.1.3 Vortex generators plus Mini Gurney flaps (NACA)
- 4.2.2 Mini Gurney flaps (DU97W300)

The following subsections were removed entirely (pls note that the numbering of the sections and Figures has changed as they refer to the previous version of this manuscript):
- Sect. 4.1.1 Baseline (NACA)
- Sect. 4.2.3: Fig. 17+text
- Appendix A and everything that had to do with the third airfoil AH97W174

**Commented [D40]:** RC1(Q18minor): the fact that the L/D increase is marginal was highlighted

[revised manuscript text omitted]

**Dear Referee 1,**

**on behalf of all co-authors, I would like to thank you for taking the time to review our study. Your suggestions will definitely improve the quality of this paper.**

**Most of your suggestions will be implemented in the revised paper. In only a few cases, the comments are contradictory to those of Referee 2 and a synthesis of the suggestions will be tried.**

**Detailed one-to-one answers can be found in bold letters in the following.**

**Best regards,**

**Jörg Alber**

General comments:

This paper present wind tunnel investigations on the effects of Mini Gurney flaps (MGFs) and their combination with vortex generators (VGs) on the performance of airfoils and wind turbine rotor blades.

This paper present high quality experiments with a lot of details on how to design the combined configurations of MGF and VG. Output results on the efficiency of passive devices should however be taken with caution because of two main reasons:

- the zztape effects are more important than passive device effects which is certainly an important issue for real blades that have generally LE erosion during operation. Same studies with different LE roughness should certainly be performed as pointed out by the authors.

- experiments were performed in a low turbulent intensity wind tunnel facility, which is far from the environment of operated blades and may lead to a decrease of actuator efficiency.

An important output of the present paper is the new design opportunities (chord length significantly reduced) that is provided using these passive devices. This study explains to do such a new blade design (the first detailed paper on that matter from my knowledge).

It has however some issues that need to be corrected. The major issues concern :

- the scientific objective that is unclear regarding the available literature of section 1.5.

- the hypothesis on the design of MGF and Vgs that are not always formulated, especially when using Xfoil to design actuators at high angle of incidence.

**These issues will be tackled in response to the comments below.**

I thus recommend the publication of the paper with corrections detailed below.

MAJOR ISSUES:

Q1: P6L137 How the present study is original from the existing literature of section 1.5 ?

**Section 1.5 is a literature review on experimental studies that investigated the effect of GFs coupled with VGs based on similar wind tunnel tests. The main differences between the present study and the given references are:**

**1.) Storms et al. (1994), NACA 4412: This not a wind turbine airfoil. Furthermore, the Gurney flap (GF) height is 1.25 %c, which is not considered to be a mini Gurney flap (MGF) according to the definition established by the authors.**

**2.) Fuglsang et al. (2003). This is the only comprehensive study that the authors could find, which is based on wind turbine airfoils (of the Risø family). However, the size of the GF is 1%c and this is not considered to be a MGF. The VG design is similar in height, but different in the spacing between the vanes, here D = 4.2%c rather than D = 7%c. Furthermore, the current wind tunnel tests are based on three wind turbine airfoils with different characteristics. In addition, our work expands on the wake interaction between both PFC devices, as compared to Fuglsang et al.**

**3.) Li-shu et al. (2013), WA251A: This not a wind turbine airfoil and, again, the GF height is 0.9 %c doesn't qualify as MGF.**

**In summary, we consider our work original because of the following points:**

**- the definition and implementation of MGFs, which are smaller than more "conventional" GFs**

**- the interaction between MGF and relatively small VGs**

**- the use of the wind tunnel data for rotor blade simulations.**

**The Sect. 1.5. will be reformulated to make these aspects clearer.**

Q2: p8L190: "relative strong turbulence intensity of Ti=0.3%"

For atmospheric flows in which wind turbine operates, the turbulence intensity is rather around

10%, please remove strong and put it in the context of wind tunnel facilities.

**This is correct. The sentence will be reformulated to clarify that this refers to a comparison with the wind tunnel conditions of other low-turbulence wind tunnels. As such, the inflow turbulence is stronger at TU Berlin compared to e.g. TU Delft.**

Q3: P9L216: this "The model revealed a non-proportional dependency on the GF height"

contradict this "diminishing HGF, dCL/hGF increased, whereas dCd/hGF decreased.

Also, from equation 6: Cl/Cd ~ dCl/Cd , which also contradicts the non-proportional dependency …

It is not clear what is non-proportional to what ?

**Agreed, the statement will be reformulated with more clarity.**

**Previous research showed that decreasing the GF height has a beneficial effect on L/D. This observation led to the basic assumption that mini GF < 0.5%c are likely to be beneficial in terms of lift and L/D for Reynolds numbers between one and two million. This assumption was validated by means of the current wind tunnel tests.**

Q4: P9L221: hGF is defined relatively to the blade chord while the conclusion is "GF needs to be

submerged deeply into the local BL"

The height of the device should be expressed relatively to the boundary layer thickness to do that

conclusion. Also, it can't be reduced to the boundary layer thickness dependency only, as the

boundary layer is never in equilibrium on blades but subjected to different pressure gradient history depending on the blade shape. The conclusion of Alber et al (2017) study is therefore limited to the tested configurations.

**The GF height is given in relation to the chord length (%c). It is crucial that the GF is significantly smaller than the BL at design conditions, i.e. at the AoA where L/D is maximal. Based on our research, the GF works best if it is between one and two times the displacement thickness ($\delta^*$), as simulated by XFOIL. For comparison, this is in the order of 0.25% the turbulent BL thickness $\delta$ (according to the 99%*free flow- definition). Please note that these indications are guidelines.**

**Yes, the BL is not static and depends on a variety of factors, such as Re, AoA, suction or pressure side, the chordwise position and the transition location. Hence, the ratio between GF and BL should be taken at the design AoA. Furthermore, it is important that the wind tunnel tests cover a wider range of AoA, at least up to $c_{l,max}$.**

**Yes, the conclusions are always limited to the tested configurations. For instance, looking at very large Reynolds numbers, $\delta^*$ is significantly smaller and so is the optimum MGF height that should be used.**

**This part will be clarified.**

Q5: P9L224: "ratio between the GF height and the BL displacement thickness at the TE"

Why at the TE?

This is rather at the location of the GF. The effect of GF location is certainly another parameter that needs to be explored.

**The BL thickness depends on the chordwise location. We found that (using XFOIL) $\delta^*$ at the position of the GF itself (i.e. close to the TE) works well in order to determine the height of the flap.**

**Yes, the chordwise position of the GF is crucial. However, this study is restricted to the "classic" position at the TE. This aspect will be highlighted with more clarity.**

Q6: P9L225: It is not clear here why MGF is designed at the optimal angle of incidence ? It is certainly the angle of incidence corresponding to the maximum TE boundary layer thickness, so the MGF design is detrimental to other angles of incidence?

Please explain.

**The rotor blades are designed in relation to the design AoA, where the aerodynamic efficiency is maximum (L/D_max) to extract maximum power. The idea is precisely that the MGF is significantly smaller than the local BL, so that it is beneficial throughout the relevant range of AoA (i.e. up to $c_{l,max}$). Based on our experimental results, this approach worked well.**

Q7: P10L2: "hMGF < delta*", the impact of MGF on the airfoil performance becomes insignificant.

Even if it seems obvious that the MGF size has some low limitation, how do you end-up with this value ?

**We have evaluated numerous experimental studies of cambered airfoils. In general, it was observed that the GF height needs to be significantly smaller than half of the turbulent BL in order to have a beneficial effect on both lift and L/D(max). The equivalent displacement thickness is in the range of $\delta^* < h_{GF} < 2\,\delta^*$.**

**According to wind tunnel tests, this XFOIL-based assumption is useful to estimate an appropriate MGF height. Yes, if the MGF becomes "too small" in relation to the local BL, its effect vanishes. Hence, it is assumed that below the displacement thickness $\delta^*$, the MGF will not be effective. However, we didn't test such tiny devices because it didn't seem practical or helpful to do so.**

**This statement will be clarified.**

Q8: p10L228: "hMGF ~ 0.25 delta"

Why chosing 1/4 while 2/3 would lead to an higher aerodynamic impact and is stil compatible to eq. 8 ?

**The design considerations refer to $\delta^*$, which is simulated by XFOIL. Additionally, $\delta^*$ can be turned into $\delta$ to get a clearer idea in terms of the 99% BL definition. For the purpose of wind tunnel tests, different GF heights are chosen to cover a plausible range of GF heights.**

Q9: p10L234: "0.1%c < hMGF < 0.7%c"

According to table 3, the minimum value of delta is 0.82%c, leading to hMGF=0.2%c (according to equation 9). Please correct.

**The definition of a MGF is provided by equation 8 ($\delta^* < h_{GF} < 2\,\delta^*$). Equation 9 was included because $\delta$ (99%) is often used in literature. However, XFOIL is only capable of calculating $\delta^*$, not $\delta$.**

**The relation between eq. 8 and 9 will be reformulated in the review to avoid misunderstandings.**

Q10: P10L239: "All tested … in relation to the size of GF", this sentence is not clear, is the GF height varying from 0.3mm to 0.6mm ? It does not seems so for the smallest MGF as the chord is c=0.6m and hMGF=0.25%c=0.15mm. Please make it clearer.

**The indications in mm (0.15mm) refer to the wall thickness of the brass profile, not the height of the GF. In this example the GF height is 0.25%c*0.6m = 1.5 mm**

**This indication will be clarified.**

Q11: P10L241: on the vortex generators design

The BL transition can be extracted from Xfoil. However, Xfoil is known to be limited to attached flow configurations with difficulties to correctly predict forces when there exist flow separation and especially at Clmax. I also don't know any Xfoil output on the mean separation line.

It is therefore not clear here how the location of the mean separation line is obtained ?

Please be clearer.

**The purpose of VGs is to delay separation. This mechanism is relevant for AoA close to cl (max). Before stall, it is preferable for the VGs to be submerged into the BL to limit the drag increase.**

**Yes, close to separation, XFOIL is less reliable. Referring to several previous studies (given as references), the δ\* calculation is considered to be sufficiently accurate for the purpose of the current wind tunnel tests.**

**It will be clarified that the design consideration of VGs are of low-order.**

**It is possible to estimate the mean separation line with XFOIL by looking at the BL of the suction side, as long as stall is not complete. The VGs are usually placed at a chordwise position closer to the leading edge, i.e. relatively far from the separation line at cl (max). Hence, the chordwise position of the laminar-to-turbulent BL transition is more relevant in terms of the VG height.**

Q12: Also, once the flow is separated, there is not anymore a boundary layer flow, so the standard

boundary layer thickness definition fails. Please specify how do you define it ?

**The Reviewer is right. The BL thickness needs to be calculated when separation is about to be initiated at cl (max), i.e. when there still is a BL (apart from the TE separation bubble).**

Q13: P11L1: "at stall, delta is similar in both the clean and tripped cases"

Stall configuration refers to a full flow separation over the blade suction side, so no delta can be

measured (as there no boundary layer anymore).

Please reformulate to be clearer.

**Agreed. This statement will be reformulated accordingly.**

Q14: P17L377: From

Dan H. Neuhart and Catherine B. McGinley "Free-Stream Turbulence Intensity in the

Langley 14- by 22-Foot Subsonic Tunnel" NASA report NASA/TP-2004-213247, Langley

Research Center, Hampton, Virginia

The turbulent intensity is one order of magnitude higher, between ~0.07% to 0.08%.

There is certainly a mistake in the reported Ti (0.005%), please correct.

**Agreed. This statement will be corrected.**

Q15: The drag signal is acquired at 10kHz and the lift is acquired at 5kHz, it is therefore possible to

plot the standard deviation with the AOA. Please add this quantity that will help to evaluate further

the actuator efficiency.

**The signal of the force balance is indeed captured at 5 kHz. However, the mean lift value was created automatically without storing the time-resolved measurements so that this quantity cannot be provided.**

**The wake rake signal consists of 60 pressure tubes, so that the StDev can only be provided for each tube individually, as already illustrated in Fig. 12.**

**Furthermore, the 2[nd] review suggests the paper to be shortened. Therefore, we would prefer not to include this information for the other airfoils, too.**

Conclusion:

The ZZ tape has more impact on the L/D ratio than the actuators themselves. Therefore authors

raise naturally the question of the blade roughness impact on their conclusion, but should also raise the question of the turbulent intensity impact, that has the ability to enhance the mixing rate of separated shear layers near the maximum lift values (or near stall).

**Agreed.**

**Unfortunately, testing different types of surface roughness is beyond the scope of this study. The aspect of turbulence intensity is surely relevant. In the wind tunnel of TU Berlin, it is currently not possible to generate reproducible inflow turbulence similar to open field conditions at the hub height of a wind turbine. Apart from that, it is common practice to measure airfoils at low-turbulent inflow.**

**Please note: due to LE roughness (here ZZ tape), the clean rotor blade suffers from the fact that the design tip speed ratio (TSR) is increased due to the loss in lift, in this case from TSR(opt) = 7 to 8 (see Fig. 23, p.32). As such, the blade is running sub-optimally, as such causing an additional drop in L/D at each blade element leading to power decrease. This lift decrease can be alleviated by adding a MGF, which enhances lift. Hence, looking at the rotating blade, the main effect of the MGF is the re-adjustment of the TSR in order to bring it back, i.e., closer, to the optimum TSR.**

MINOR ISSUES:

Q1: This citation is not a peer review journal nor a conference paper:

Schatz, M., Gunther, B., and Thiele, F.: Numerical Simulation of the Unsteady Wake behind

Gurney-Flaps, available at:

https://www.cfd.tu-berlin.de/research/flowcontrol/gurneys_en/ (last access: 17 October 2020),

2004a.

**Agreed. The images are identical to the peer reviewed article (see Schatz et al. 2014). In order to avoid copyright issue with AIAA, we used the freely available images from our university website, which are identical.**

Q2: L93: "the unsteadiness vanished as the AoA is increased from 0 to 4°"

why only until 4° ? It increases again afterwards ?

**No, according to this prior study at TU Berlin of 2014, the additional unsteadiness of the MGF (compared to the baseline airfoil) vanishes at 4°. This means that for AoA > 4°, the additional unsteadiness due to the MGF was of minor importance or irrelevant.**

**This statement will be clarified.**

Q3: P2L37: what PFC stands for ?

**Passive flow control (PFC). The abbreviation will be included.**

Q4: P12L267: what is the purpose of the "airfoil box" ?

**At TU Berlin, we use solid metal structures, so-called boxes, that can be lifted into the test section of the wind tunnel. The airfoil box is specific for testing different airfoil sections (wings).**

**The term will be explained with more clarity.**

Q5: P12L274: from the static pressure difference between the inlet and outlet of a duct you only get

the pressure losses, not the dynamic pressure used to measurement the inflow velocity. Please correct the sentence.

**There is a ring line of pressure tabs at both the inlet and the outlet of the duct determining the static pressure. The flow velocity can be calculated by the static pressure difference and the contraction ration (according to Bernoulli for horizontal and incompressible air flows):**

$$v_2^2 = \frac{2 \cdot (p1 - p2)}{\rho - \rho \left(\frac{A2}{A1}\right)^2}$$

**where v2 is the flow velocity in the test section and p1 the integrated static pressure at the inlet and p2 at the outlet.**

**The measurement principle will be explained in more depth.**

Q6: P12L276: How the blade is attached to the balance system ?

**The airfoil model is decoupled form the wind tunnel and fixed directly to the metal beam of the force balance.**

**The installation principle will be clarified.**

Q7: P12L277: What are the wind tunnel boundary layer thickness value ? And what are the end plates dimensions and how they were chosen ?

**The boundary layer thickness at the wind tunnel walls were not measured during the tests.**

**The dimension of the end plates will be included.**

Q8: P12L277-278: The flow in the bypass region is certainly very complex with interaction between the facility boundary-layer and the end plate boundary layer. Why using that location as a reference velocity ? Is the prandtl tube in front of the blade sufficiently insufficient ? Is there blade induction effects at this location ?

**The Prandtl tube in front of the airfoil model is only a reference in order to validate the inflow velocity, see Q5: P12L274. For that, the flow velocity is measured without the airfoil model and compared between the ring line and the Prandtl tube.**

**The Prandtl tube will be removed from the Figure to avoid misunderstandings.**

Q9: P14L1: what is the origin of x ? because x=c from figure 4a is at the trailing edge … I guess you mean x/c=2 ?

Please correct.

**This indication refers to the distance between the airfoil trailing edge and the pressure tubes of the wake rake. Hence, the wake rake is positioned exactly 1 chord-length, i.e. 0.6m or 100 %c, behind the airfoil trailing edge.**

**This indication will be specified with more clarity.**

Idem p25L494: 100%c → x/c=2

**Again, this means that the distance between the airfoil and the rake is 1 chord length or 100 %c.**

**This indication will be specified with more clarity.**

Q10: P14L2: what do you mean by "return to static pressure level in the wind tunnel" ?

If the pressure is measured in the blade wake area at high angle of incidence, 1D is certainly not

sufficient to return to the static pressure level in the wind tunnel. Please be more specific.

**According to the references given in the report (Barlow et al. p.178), the wake rake should be located at least 0.7c behind trailing edge for the static pressure not to be affected by the unsteadiness of the airfoil wake.**

**This statement will be specified.**

Q11: P14L311: "two Prandtl tubes that are installed inside the downstream plane of the rake, one on

top and one below the casing"

This sentence is not clear as Prandtl tubes measure the dynamic pressure (difference between static

and total pressure). Please be clearer.

**Prandtl tubes are used to measure both the total pressure (in front) and the static pressure (from the side). This method will be clarified.**

Q12: P14: careful on the notations: deltapP(yi)=deltaPi of equation 11?

Ptotal = Pbar total (the question behind is: how long the signal was acquired and does the bar mean

averaged over that signal ?)

Pstatic=Pbar static (idem)?

**Yes, the bar on top of the p indicates that each signal (of each pressure tube) was averaged over the measurement duration of 5 sec per AoA (after 4 sec buffer).**

**This indication will be clarified.**

Q13: P14L330: please explain (quickly) where the relation 14 comes from.

**Eq. 14 is the practical and detailed implementation of the momentum loss, see e.g. Barlow et al. p. 177.**

[Figure]

$$c_{d0} = 2\int \left(\sqrt{\frac{q}{q_0}} - \frac{q}{q_0}\right)\frac{dy}{c}$$

Q14: P14L340: About Lambda: is it related to the airfoil thickness (and so dependent on each

airfoil), or is it the height of the test section (and so a fixe value). Please make it clearer.

**Lambda is the so-called body shape factor as introduced by Allen and Vincenti (1944). It is a function of the maximum thickness of the aerodynamic body, here the airfoil model. For brevity, the reader is referred to secondary literature, such as Barlow et al. p. 352.**

Q15: Please provide the raw lift and drag curves to see the improvement when applying the wind

tunnel correction.

**The raw data would probably not provide any additional insight. For space economy and according to the 2nd review (length of the report), we would prefer to not include more data.**

Q16: P18L396: Next, figure 12a … isn't it figure 11a ?

Please check.

**Yes, this mistake will be corrected.**

Q17: P18 legend of figure 11 on p19.

Please correct

**Agreed. The layout will be corrected.**

Q18: p21L1-2: it should also be noted that there is only marginal L/D improvement between GF cases.

**The marginal L/D improvement in the clean case of Fig. 13 (b) will be highlighted.**

Q19: P26L506: normalizing the vertical position with the chord and locate the center relatively to the blade will help to evaluate the wake extend and thus conclude on the flow separation extend, that is not necessarily going up to the trailing edge - stalled (in figure 15a: the baseline zz case indicates that the flow separation do not occurs at the leading edge).

**The Reviewer's suggestion is not completely clear to us. The vertical positions of the rake tubes are normalized by the total rake span, i.e. the distance from the top to the bottom tube. Hence, y = 0.0 is the exact middle of the wake rake. At positive AoA, the minimum pressure point is pushed towards the wind tunnel floor due to the downwash effect in the airfoil wake.**

**Using ZZ tape, Fig. 15 (a) indicates that stall is happening without the drop in lift in the clean case. This is a common observation of airfoil measurements in wind tunnels.**

Q20: P26L509: please replace "suppress" by "delay"

**Okay.**

Q21: p27 Figure 19: I don't see the benefit of adding GF (or MGF) compared to VG alone, and it is not commented in the article.

Please explain why MGF+VG is better than VG alone for that configuration ?

**The benefit of using both MGF+VG is that the L/D ratio is slightly improved and, more importantly, that pre-stall lift is significantly elevated. The latter aspect is crucial in order to mitigate the lift decrease due to roughness on the rotating blade.**

**This benefit will be highlighted.**

Q22: P30: legend of figure 20 is misplaced.

**The legend will be corrected.**

Dear Mohammad Mahfouz (RC2),

on behalf of all co-authors, I would like to thank you for taking the time to review our study. Your suggestions will definitely improve the quality of this paper.

Most of your suggestions will be implemented in the revised paper. The report will be restructured and shortened, as suggested. In only a few cases, your comments cannot be implemented entirely.

Detailed one-to-one answers are provided inside the pdf supplement.

Best regards,

Jörg Alber

[revised manuscript text omitted]